# Caveats for information bottleneck in deterministic scenarios

**Artemy Kolchinsky & Brendan D. Tracey**[*]
Santa Fe Institute
Santa Fe, NM 87501, USA
{artemyk,tracey.brendan}@gmail.com

**Steven Van Kuyk**
School of Engineering and Computer Science
Victoria University of Wellington, New Zealand
steven.jvk@gmail.com

## Abstract

Information bottleneck (IB) is a method for extracting information from one random variable $X$ that is relevant for predicting another random variable $Y$. To do so, IB identifies an intermediate "bottleneck" variable $T$ that has low mutual information $I(X;T)$ and high mutual information $I(Y;T)$. The *IB curve* characterizes the set of bottleneck variables that achieve maximal $I(Y;T)$ for a given $I(X;T)$, and is typically explored by maximizing the *IB Lagrangian*, $I(Y;T) - \beta I(X;T)$. In some cases, $Y$ is a deterministic function of $X$, including many classification problems in supervised learning where the output class $Y$ is a deterministic function of the input $X$. We demonstrate three caveats when using IB in any situation where $Y$ is a deterministic function of $X$: (1) the IB curve cannot be recovered by maximizing the IB Lagrangian for different values of $\beta$; (2) there are "uninteresting" trivial solutions at all points of the IB curve; and (3) for multi-layer classifiers that achieve low prediction error, different layers cannot exhibit a strict trade-off between compression and prediction, contrary to a recent proposal. We also show that when $Y$ is a small perturbation away from being a deterministic function of $X$, these three caveats arise in an approximate way. To address problem (1), we propose a functional that, unlike the IB Lagrangian, can recover the IB curve in all cases. We demonstrate the three caveats on the MNIST dataset.

## 1 Introduction

The *information bottleneck* (IB) method (Tishby et al., 1999) provides a principled way to extract information that is present in one variable that is relevant for predicting another variable. Given two random variables $X$ and $Y$, IB posits a "bottleneck" variable $T$ that obeys the Markov condition $Y - X - T$. By the data processing inequality (DPI) (Cover & Thomas, 2012), this Markov condition implies that $I(X;T) \geq I(Y;T)$, meaning that the bottleneck variable cannot contain more information about $Y$ than it does about $X$. In fact, any particular choice of the bottleneck variable $T$ can be quantified by two terms: the mutual information $I(X;T)$, which reflects how much $T$ compresses $X$, and the mutual information $I(Y;T)$, which reflects how well $T$ predicts $Y$. In IB, bottleneck variables are chosen to maximize prediction given a constraint on compression (Witsenhausen & Wyner, 1975; Ahlswede & Körner, 1975; Gilad-Bachrach et al., 2003),

$$F(r) := \max_{T \in \Delta} I(Y;T) \quad \text{s.t.} \quad I(X;T) \leq r, \tag{1}$$

where $\Delta$ is the set of random variables $T$ obeying the Markov condition $Y - X - T$. The values of $F(r)$ for different $r$ specify the *IB curve*. In order to explore the IB curve, one must find optimal $T$ for different values of $r$. It is known that the IB curve is concave in $r$ but may not be *strictly* concave. This seemingly minor issue of non-strict concavity will play a central role in our analysis.

In practice, the IB curve is almost always explored not via the constrained optimization problem of Eq. (1), but rather by maximizing the so-called *IB Lagrangian*,

$$\mathcal{L}_{\text{IB}}^{\beta}(T) := I(Y;T) - \beta I(X;T), \tag{2}$$

---

[*]Dept of Aeronautics & Astronautics, Massachusetts Institute of Technology, Cambridge, MA 02139, USA

where $\beta \in [0, 1]$ is a parameter that controls the trade-off between compression and prediction. The advantage of optimizing $\mathcal{L}_{\text{IB}}^{\beta}$ is that it avoids the non-linear constraint in Eq. (1)[1].

Several recent papers have drawn connections between IB and *supervised learning*, in particular classification using neural networks. In this context, $X$ represents input vectors, $Y$ represents the output classes, and $T$ represents intermediate representations used by the network architecture, such as the activity of hidden layer(s) (Tishby & Zaslavsky, 2015). Some of these papers modify neural network training algorithms so as to optimize the IB Lagrangian (Alemi et al., 2016; Chalk et al., 2016; Kolchinsky et al., 2017), thereby permitting the use of IB with high-dimensional, continuous-valued random variables. Some papers have also suggested that by controlling the amount of compression, one can tune desired characteristics of trained models such as generalization error (Shamir et al., 2010; Tishby & Zaslavsky, 2015; Vera et al., 2018), robustness to adversarial inputs (Alemi et al., 2016), and detection of out-of-distribution data (Alemi et al., 2018). Other research (Shwartz-Ziv & Tishby, 2017) has suggested — somewhat controversially (Saxe et al., 2018) — that stochastic gradient descent (SGD) training dynamics may implicitly favor hidden layer mappings that balances compression and prediction, with earlier hidden layers favoring prediction over compression and latter hidden layers favoring compression over prediction. Finally, there is the general notion that intermediate representations that are optimal in the IB sense correspond to "interesting" or "useful" compressions of input vectors (Amjad & Geiger, 2018).

There are also numerous application domains of IB beyond supervised learning, including clustering (Slonim & Tishby, 2000), coding theory and quantization (Cardinal, 2003; Zeitler et al., 2008; Courtade & Wesel, 2011), and cognitive science (Zaslavsky et al., 2018). In most of these applications, it is of central interest to explore solutions at different points on the IB curve, for example to control the number of detected clusters, or to adapt codes to available channel capacity.

In some scenarios, $Y$ may be a deterministic function of $X$, i.e., $Y = f(X)$ for some single-valued function $f$. For example, in many classification problems, it is assumed that any given input belongs to a single class, which implies a deterministic relationship between $X$ and $Y$. In this paper, we demonstrate three caveats for IB that appear whenever $Y$ is a deterministic function of $X$:

1. There is no one-to-one mapping between different points on the IB curve and maximizers of the IB Lagrangian $\mathcal{L}_{\text{IB}}^{\beta}$ for different $\beta$, thus the IB curve cannot be explored by maximizing $\mathcal{L}_{\text{IB}}^{\beta}$ while varying $\beta$. This occurs because when $Y = f(X)$, the IB curve has a piecewise linear shape and is therefore not strictly concave. The dependence of the IB Lagrangian on the strict concavity of $F(r)$ has been previously noted (Gilad-Bachrach et al., 2003; Shwartz-Ziv & Tishby, 2017), but the implications and pervasiveness of this problem (e.g., in many classification scenarios) has not been fully recognized. We analyze this issue and propose a solution in the form of an alternative objective function, which can be used to explore the IB curve even when $Y = f(X)$.
2. All points on the IB curve contain uninteresting trivial solutions (in particular, stochastic mixtures of two very simple solutions). This suggests that IB-optimality is not sufficient for an intermediate representation to be an interesting or useful compression of input data.
3. For a neural network with several hidden layers that achieves a low probability of error, the hidden layers cannot display a strict trade-off between compression and prediction (in particular, different layers can only differ in the amount of compression, not prediction).

In Appendix B, we show that the above three caveats also apply to the recently proposed *deterministic IB* variant of IB (Strouse & Schwab, 2017), in which the compression term is quantified using the entropy $H(T)$ rather than the mutual information $I(X; T)$. In that Appendix, we propose an alternative objective function that can be used to resolve the first problem for dIB.

We also show, in Appendix C, that our results apply when $Y$ is not exactly a deterministic function of $X$, but $\epsilon$-close to one. In this case: (1) it is hard to explore the IB curve by optimizing the IB Lagrangian, because all optimizers will fall within $\mathcal{O}(-\epsilon \log \epsilon)$ of a single "corner" point on the information plane; (2) along all points on the IB curve, there are "uninteresting" trivial solutions that are no more than $\mathcal{O}(-\epsilon \log \epsilon)$ away from being optimal; (3) different layers of a neural networks can trade-off at most $\mathcal{O}(-\epsilon \log \epsilon)$ amount of prediction.

A recent paper (Amjad & Geiger, 2018) also discusses several difficulties in using IB to analyze intermediate representations in supervised learning. That paper does not consider the particular

---

[1]Note that optimizing $\mathcal{L}_{\text{IB}}^{\beta}$ is still a constrained problem in that $p(t|x)$ must be a valid conditional probability. However, this constraint is usually easier to handle, e.g., by using an appropriate parameterization.

issues that arise when $Y$ is a deterministic function of $X$, and its arguments are complementary to ours. Shwartz-Ziv & Tishby (2017)[Sec. 2.4] discuss another caveat for IB in deterministic settings, concerning the relationship between sufficient statistics and the complexity of the input-output mapping, which is orthogonal to the three caveats analyzed here. Finally, Saxe et al. (2018) and Amjad & Geiger (2018) observed that when $T$ is continuous-valued and a deterministic function of a continuous-valued $X$, $I(X;T)$ can be unbounded, making the application of the IB framework problematic. We emphasize that the caveats discussed in this paper are unrelated to that problem.

Note that our results are based on analytically-provable properties of the IB curve, i.e., global optima of Eq. (1), and do not concern practical issues of optimization (which may be important in real-world scenarios). Our theoretical results are also independent of the practical issue of estimating MI between neural network layers, an active area of recent research (Belghazi et al., 2018; Goldfeld et al., 2018; Dai et al., 2018; Gabrié et al., 2018), though our empirical experiments in Section 7 rely on the estimator proposed in (Kolchinsky & Tracey, 2017; Kolchinsky et al., 2017). Finally, our results are also independent of issues related to the relationship between IB, finite data sampling, and generalization error (Shamir et al., 2010; Tishby & Zaslavsky, 2015; Vera et al., 2018).

In the next section, we review some of the connections between supervised learning and IB. In Section 3, we show that when $Y = f(X)$, the IB curve has a piecewise linear (not strictly concave) shape. In Sections 4, 5 and 6, we discuss the three caveats mentioned above. In Section 7, we demonstrate the caveats using a neural-network implementation of IB on the MNIST dataset.

## 2 SUPERVISED CLASSIFICATION AND IB

In supervised learning, one is given a training dataset $\{x_i, y_i\}_{i=1..N}$ of inputs and outputs. In this case, the random variables $X$ and $Y$ refer to the inputs and outputs respectively, and $\bar{p}(x, y)$ indicates their joint empirical distribution in the training data. It is usually assumed that the $x$'s and $y$'s are sampled i.i.d. from some "true" distribution $w(y|x)w(x)$. The high-level goal of supervised learning is to use the dataset to select a particular conditional distribution $q_\theta(\hat{y}|x)$ of outputs given inputs parameterized by $\theta$ that is a good approximation of $w(y|x)$. For clarity, we use $\hat{Y}$ (and $\hat{y}$) to indicate a random variable (and its outcome) corresponding to the *predicted* outputs. We use $\mathcal{X}$ to indicate the set of outcomes of $X$, and $\mathcal{Y}$ to indicate the set of outcomes of $Y$ and $\hat{Y}$. Supervised learning is called *classification* when $Y$ takes a finite set of values, and *regression* when $Y$ is continuous-valued. Here we focus on classification, where the set of possible output values can be written as $\mathcal{Y} = \{0, 1, \ldots, m\}$. We expect our results to also apply in some regression scenarios (with some care regarding evaluating MI measures, see Footnote 2), but leave this for future work.

In practice, many supervised learning architectures use some kind of intermediate representation to make predictions about the output, such as hidden layers in neural networks. In this case, the random variable $T$ represents the activity of some particular hidden layer in a neural network. Let $T$ be a (possibly stochastic) function of the inputs, as determined by some parameterized conditional distribution $q_\theta(t|x)$, so that $T$ obeys the Markov condition $Y - X - T$. The mapping from inputs to hidden layer activity can be either deterministic, as traditionally done in neural networks, or stochastic, as used in some architectures (Alemi et al., 2016; Kolchinsky et al., 2017; Achille & Soatto, 2018)[2]. The mapping from hidden layer activity to predicted outputs is represented by the parameterized conditional distribution $q_\theta(\hat{y}|t)$, so the full Markov condition between true outputs, inputs, hidden layer activity, and predicted outputs is $Y - X - T - \hat{Y}$. The overall mapping from inputs to predicted outputs implemented by a neural network with a hidden layer can be written as

$$q_\theta(\hat{y}|x) := \int q_\theta(\hat{y}|t)q_\theta(t|x)dt \,. \tag{3}$$

Note that $T$ does not have to represent a particular hidden layer, but could instead represent the activity of the neurons in a set of contiguous hidden layers, or in fact any arbitrary set of neurons that separate inputs from predicted outputs (in the sense of conditional independence, so that Eq. (3) holds). More generally yet, $T$ could also represent some intermediate representation in a non-neural-network supervised learning architecture, again as long as Eq. (3) holds.

---

[2]If $T$ is continuous-valued and a deterministic function of a continuous-valued $X$, then one should consider noisy or quantized mappings from inputs to hidden layers, such that $I(X;T)$ is finite.

For classification problems, training often involves selecting $\theta$ to minimize *cross-entropy loss*, sometimes while stochastically sampling the activity of hidden layers. When $T$ is the last hidden layer, or when the mapping from $T$ to the last hidden layer is deterministic, cross-entropy loss can be written

$$\mathcal{L}_{\text{CE}}(\theta) = -\frac{1}{N}\sum_i \int q_\theta(t|x_i)\log q_\theta(\hat{y}_i|t)\,dt = \mathbb{E}_{\bar{p}_\theta(Y,T)}\left[-\log q_\theta(\hat{Y}|T)\right], \qquad (4)$$

where $\mathbb{E}$ indicates expectation, $\bar{p}_\theta(y,t) := \frac{1}{N}\sum_i \delta(y,y_i)q_\theta(t|x_i)$ is the empirical distribution of outputs and hidden layer activity, and $\delta$ is the Kronecker delta. Eq. (4) can also be written as

$$\mathcal{L}_{\text{CE}}(\theta) = H(\bar{p}_\theta(Y|T)) + D_{\text{KL}}(\bar{p}_\theta(Y|T)\|q_\theta(\hat{Y}|T)) \qquad (5)$$

where $H$ is (conditional) Shannon entropy, $D_{\text{KL}}$ is Kullback-Leibler (KL) divergence, and $\bar{p}_\theta(y|t)$ is defined via the definition of $\bar{p}_\theta(y,t)$ above. Since KL is non-negative, $\mathcal{L}_{\text{CE}}(\theta)$ is an upper bound on the conditional entropy $H(\bar{p}_\theta(Y|T))$. During training, one can decrease $\mathcal{L}_{\text{CE}}$ either by decreasing the conditional entropy $H(\bar{p}_\theta(Y|T))$, or decreasing the KL term by changing $q_\theta(\hat{Y}|T)$ to better approximate $\bar{p}_\theta(Y|T)$. Minimizing $\mathcal{L}_{\text{CE}}$ thus minimizes an upper bound on $H(\bar{p}_\theta(Y|T))$, which is equivalent to maximizing a lower bound on $I_{\bar{p}_\theta}(Y;T) = H(\bar{p}(Y)) - H(\bar{p}_\theta(Y|T))$ (since $H(\bar{p}(Y))$ is a constant that doesn't depend on $\theta$).

We can now make explicit the relationship between supervised learning and IB. In IB, one is given a joint distribution $p(x,y)$, and then seeks a bottleneck variable $T$ that obeys the Markov condition $Y - X - T$ and minimizes $I(X;T)$ while maximizing $I(Y;T)$. In supervised learning, one is given an empirical distribution $\bar{p}(x,y)$ and defines an intermediate representation $T$ that obeys the Markov condition $Y - X - T$; during training, cross-entropy loss is minimized, which is equivalent to maximizing a lower bound on $I(Y;T)$ (given assumptions of Eq. (4) hold). To complete the analogy, one might choose $\theta$ so as to also minimize $I(X;T)$, i.e., choose hidden layer mappings that provide compressed representations of input data (Alemi et al., 2016; Chalk et al., 2016; Kolchinsky et al., 2017; Achille & Soatto, 2018). In fact, we use such an approach in our experiments on the MNIST dataset in Section 7.

In many supervised classification problems, there is only one correct class label associated with each possible input. Formally, this means that given the empirical training dataset distribution $\bar{p}(y|x)$, one can write $Y = f(X)$ for some single-valued function $f$. This relationship between $X$ and $Y$ may arise because it holds under the "true" distribution $w(y|x)$ from which the training dataset is sampled, or simply because each input vector $x$ occurs at most once in the training data (as happens in most real-world classification training datasets). Note that we make no assumptions about the relationship between $X$ and $\hat{Y}$ under $q_\theta(\hat{y}|x)$, the distribution of predicted outputs given inputs implemented by the supervised learning architecture, and our analysis holds even if $q_\theta(\hat{y}|x)$ is non-deterministic (e.g., the softmax function).

It is important to note that not all classification problems are deterministic. The map from $X$ to $Y$ may be intrinsically noisy or non-deterministic (e.g., if human labelers cannot agree on a class for some $x$), or noise may be intentionally added as a regularization technique (e.g., as done in the *label smoothing* method (Szegedy et al., 2016)). Moreover, one of the pioneering papers on the relationship between IB and supervised learning (Shwartz-Ziv & Tishby, 2017) analyzed an artificially-constructed classification problem in which $p(y|x)$ was explicitly defined to be noisy (see their Eq. 10). However, we believe that many if not most real-world classification problems do in fact exhibit a deterministic relation between $X$ and $Y$. In addition, as we show in Appendix C, even if the relationship between $X$ and $Y$ is not perfectly deterministic but close to being so, then the three caveats discussed in this paper still apply in an approximate sense.

## 3 THE IB CURVE WHEN $Y$ IS A DETERMINISTIC FUNCTION OF $X$

Consider the random variables $X, Y$ where $Y$ is discrete-valued and a deterministic function of $X$, i.e., $Y = f(X)$. $I(Y;T)$ can be upper bounded as $I(Y;T) \leq I(X;T)$ by the DPI, and $I(Y;T) \leq H(Y)$ by basic properties of MI (Cover & Thomas, 2012). To visualize these bounds, as well as other results of our analysis, we use so-called "information plane" diagrams (Tishby et al., 1999). The information plane represents various possible bottleneck variables in terms of their compression ($I(X;T)$, horizontal axis) and prediction ($I(Y;T)$, vertical axis). The two bounds mentioned above are plotted on an information plane in Fig. 1. In this section, we show that when $Y = f(X)$, the IB curve saturates both bounds, and is not strictly concave but rather piece-wise linear.

To show that the IB curve saturates the DPI bound $I(Y;T) \leq I(X;T)$ for $I(X;T) \in [0, H(Y)]$, let $B_\alpha$ be a Bernoulli random variable that is equal to 1 with probability $\alpha$. Then, define a manifold of bottleneck variables $T_\alpha$ parameterized by $\alpha \in [0, 1]$,

$$T_\alpha := B_\alpha \cdot Y = B_\alpha \cdot f(X). \tag{6}$$

Thus, $T_\alpha$ is equal to $Y$ with probability $\alpha$, and equal to 0 with probability $1 - \alpha$. From Eq. (6), it is clear that $T_\alpha$ can be written either as a (stochastic) function of $X$ or a (stochastic) function of $Y$, thus both Markov conditions $Y - X - T_\alpha$ and $X - Y - T_\alpha$ hold. Applying the DPI to both Markov conditions implies that $I(X;T_\alpha) = I(Y;T_\alpha)$ for any $T_\alpha$. Note that for $\alpha = 1$, $T_\alpha = Y$ and thus $I(Y;T_\alpha) = H(Y)$, while for $\alpha = 0$, $T_\alpha = 0$

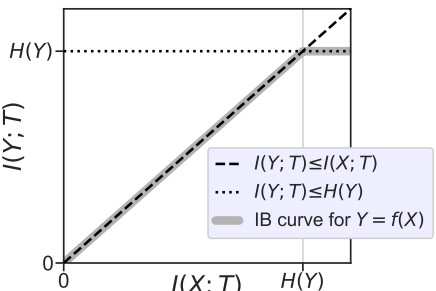

Figure 1: A schematic IB curve. Dashed line is DPI, dotted line is $I(Y;T) \leq H(Y)$. When $Y = f(X)$, the IB curve saturates both bounds.

and thus $I(Y;T_\alpha) = 0$. Since MI is continuous in probabilities, the manifold of bottleneck variables $T_\alpha$ sweeps the full range of values $\langle I(X;T_\alpha), I(Y;T_\alpha) \rangle = \langle 0, 0 \rangle$ to $\langle I(X;T_\alpha), I(Y;T_\alpha) \rangle = \langle H(Y), H(Y) \rangle$ as $\alpha$ ranges from 0 to 1, while obeying $I(X;T_\alpha) = I(Y;T_\alpha)$. Thus, the manifold of $T_\alpha$ bottleneck variables achieves the DPI bound over $I(X;T) \in [0, H(Y)]$.

Given its definition in Eq. (1), the IB curve is monotonically increasing. Since $\langle H(Y), H(Y) \rangle$ is on the IB curve (e.g., $T_\alpha$ for $\alpha = 1$), it must be that $I(Y;T) \geq H(Y)$ when $I(X;T) \geq H(Y)$ for optimal $T$. At the same time, it is always the case that $I(Y;T) \leq H(Y)$, by basic properties of MI. Thus, the IB curve is a flat line at $I(Y;T) = H(Y)$ for $I(X;T) \geq H(Y)$.

Thus, when $Y = f(X)$, the IB curve is piecewise linear and therefore not strictly concave.[3]

## 4 ISSUE 1: IB CURVE CANNOT BE EXPLORED USING THE IB LAGRANGIAN

We now show that when $Y = f(X)$, one cannot explore the IB curve by solving $\arg\max_{T \in \Delta} \mathcal{L}_{\text{IB}}^\beta$ while varying $\beta$, because there is no one-to-one mapping between points on the IB curve and optimizers of $\mathcal{L}_{\text{IB}}^\beta$ for different $\beta$. This issue is diagrammed in Fig. 2, and explained formally below.

Assume $Y = f(X)$. For any bottleneck variable $T$ and all $\beta \in [0, 1]$,[4] $\mathcal{L}_{\text{IB}}^\beta$ is upper bounded by

$$\mathcal{L}_{\text{IB}}^\beta(T) = I(Y;T) - \beta I(X;T) \leq (1 - \beta)I(Y;T) \leq (1 - \beta)H(Y) \tag{7}$$

where the first inequality uses the DPI, and the second $I(Y;T) \leq H(Y)$.

Now consider the bottleneck variable $T_{\text{copy}} := f(X) = Y$ (or, equivalently, any one-to-one transformation of $T_{\text{copy}}$), which corresponds to $T_\alpha$ for $\alpha = 1$ in Eq. (6), and is the "corner point" of the IB curve in Fig. 1. Observe that $\mathcal{L}_{\text{IB}}^\beta(T_{\text{copy}}) = (1 - \beta)H(Y)$, so $T_{\text{copy}}$ achieves the bound of Eq. (7). Thus, $T_{\text{copy}}$ optimizes $\mathcal{L}_{\text{IB}}^\beta$ for all $\beta \in [0, 1]$, though it represents only a single point on the IB curve.

The above argument shows that a single point on the IB curve optimizes $\mathcal{L}_{\text{IB}}^\beta$ for many different $\beta$. Conversely, many points on the curve can simultaneously optimize $\mathcal{L}_{\text{IB}}^\beta$ for a single $\beta$. In particular, for $\beta = 1$, $\mathcal{L}_{\text{IB}}^\beta(T) = I(Y;T) - I(X;T) \leq 0$ (the inequality follows from the DPI). Any $T$ that has $I(X;T) = I(Y;T)$ — such as the manifold of $T_\alpha$ defined in Eq. (6) — achieves the maximum $\mathcal{L}_{\text{IB}}^\beta(T) = 0$ for $\beta = 1$, so all points on the non-flat part of IB curve in Fig. 1 simultaneously maximize $\mathcal{L}_{\text{IB}}^\beta$ for this $\beta$. For $\beta = 0$, $\mathcal{L}_{\text{IB}}^\beta(T) = I(Y;T) \leq H(Y)$, and this upper bound is achieved by all $T$ that have $I(Y;T) = H(Y)$. Thus, all solutions on the flat part of the IB curve in Fig. 1 simultaneously maximize $\mathcal{L}_{\text{IB}}^\beta$ for $\beta = 0$. Note that common supervised learning algorithms that minimize cross-entropy loss can be considered to have $\beta = 0$, where there is no explicit benefit to compression, and their intermediate representations will typically fall into this regime.

The problem is illustrated visually in Fig. 2, which shows two hypothetical IB curves (in thick gray lines): a not strictly concave (piecewise linear) curve (left), as occurs when $Y$ is a deterministic

---

[3]The IB curve may be not strictly concave in other cases as well. A sufficient condition for the IB curve to be strictly concave is for $p(y|x) > 0$ for all $x, y$ (Gilad-Bachrach et al., 2003, Lemma 6).

[4]It is sufficient to consider $\beta \in [0, 1]$ because for $\beta < 0$, uncompressed solutions like $T = X$ are maximizers of $\mathcal{L}_{\text{IB}}^\beta$, while for $\beta > 1$, $\mathcal{L}_{\text{IB}}^\beta$ is non-positive and trivial solutions such as $T = \text{const}$ are maximizers.

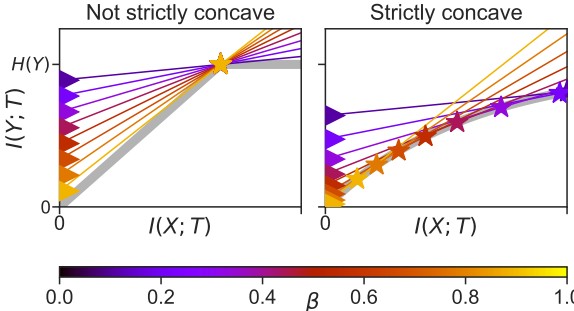

Figure 2: **Failure of IB Lagrangian.** Thick gray lines shows two possible IB curves, thin lines are isolines of $\mathcal{L}_{\text{IB}}^{\beta}$ for different $\beta$ (colors indicate different $\beta$), stars indicate achievable $\langle I(X;T), I(Y;T) \rangle$ that maximize $\mathcal{L}_{\text{IB}}^{\beta}$. For not strictly concave IB curve, different $\beta \in (0,1)$ only recover a single point. For strictly concave IB curve, different $\beta$ recover different points.

function of $X$, and a strictly concave curve (right). In both subplots, the colored lines indicate isolines of the IB Lagrangian. The highest point on the IB curve crossed by a given colored line is marked by a star, and represents the $\langle I(X;T), I(Y;T) \rangle$ values for a $T$ which optimizes $\mathcal{L}_{\text{IB}}^{\beta}$ for the corresponding $\beta$. For the piecewise linear IB curve, all $\beta \in (0,1)$ only recover a single solution ($T_{\text{copy}}$). For the strictly concave IB curve, different $\beta$ values recover different solutions.

To summarize, when $Y = f(X)$, the IB curve is piecewise linear and cannot be explored by optimizing $\mathcal{L}_{\text{IB}}^{\beta}$ while varying $\beta$. On the flip side, if $Y = f(X)$ and one's goal is precisely to find the corner point $\langle H(Y), H(Y) \rangle$ (maximum compression at no prediction loss), then our results suggest that the IB Lagrangian provides a robust way to do so, being invariant to the particular choice of $\beta$. Moreover, if one does want to recover solutions at different points on the IB curve, and $f : \mathcal{X} \to \mathcal{Y}$ is fully known, our results show how to do so in a closed-form way (via $T_{\alpha}$), without any optimization.

Is there a single procedure that can be used to explore the IB curve in all cases, whether it is strictly convex or not? In principle, this can be done by solving the constrained optimization of Eq. (1) for different values of $r$. In practice, however, solving this constrained optimization problem — even approximately — is far from a simple task, in part due to the non-linear constraint on $I(X;T)$, and many off-the-shelf optimization tools cannot handle this kind of problem. It is desirable to have an unconstrained objective function that can be used to explore any IB curve. For this purpose, we propose an alternative objective function, which we call the *squared-IB functional*,

$$\mathcal{L}_{\text{sq-IB}}^{\beta}(T) := I(Y;T) - \beta I(X;T)^2 . \tag{8}$$

We show that maximizing $\mathcal{L}_{\text{sq-IB}}^{\beta}$ while varying $\beta$ recovers the IB curve, whether it is strictly concave or not. To do so, we first analyze why the IB Lagrangian fails when $Y = f(X)$ (or, more generally, when the IB curve is piecewise linear). Over the increasing region of the IB curve, the inequality constraint in Eq. (1) can be replaced by an equality constraint (Witsenhausen & Wyner, 1975, Thm. 2.5), so maximizing $\mathcal{L}_{\text{IB}}^{\beta}$ can be written as $\max_T I(Y;T) - \beta I(X;T) = \max_r F(r) - \beta r$ (i.e., the Legendre transform of $-F(r)$). The derivative of $F(r) - \beta r$ is zero when $F'(r) = \beta$, and any point on the IB curve that has $F'(r) = \beta$ will maximize $\mathcal{L}_{\text{IB}}^{\beta}$ for that $\beta$. When $Y = f(X)$, all points on the increasing part of the curve have $F'(r) = 1$, and will all simultaneously maximize $\mathcal{L}_{\text{IB}}^{\beta}$ for $\beta = 1$. More generally, all points on a linear segment of a piecewise linear IB curve will have the same $F'(r)$, and will all simultaneously maximize $\mathcal{L}_{\text{IB}}^{\beta}$ for the corresponding $\beta$.

Using a similar argument as above, we write $\max_T \mathcal{L}_{\text{sq-IB}}^{\beta} = \max_r F(r) - \beta r^2$. The derivative of $F(r) - \beta r^2$ is 0 when $F'(r)/(2r) = \beta$. Since $F$ is concave, $F'(r)$ is decreasing in $r$, though it is not strictly decreasing when $F$ is not strictly concave. $F'(r)/(2r)$, on the other hand, is strictly decreasing in $r$ in all cases, thus there can be only one $r$ such that $F'(r)/(2r) = \beta$ for a given $\beta$. For this reason, for any IB curve, there can be only one point that maximizes $\mathcal{L}_{\text{sq-IB}}^{\beta}$ for a given $\beta$.[5]

Note also that any $r$ that satisfies $F'(r) = \beta$ also satisfies $F'(r)/(2r) = \beta/(2r)$. Thus, any point that maximizes $\mathcal{L}_{\text{IB}}^{\beta}$ for a given $\beta$ also maximizes $\mathcal{L}_{\text{sq-IB}}^{\beta}$ under the transformation $\beta \mapsto \beta/(2 \cdot I(X;T))$, and vice versa. Importantly, unlike for $\mathcal{L}_{\text{IB}}^{\beta}$, there can be non-trivial maximizers of $\mathcal{L}_{\text{sq-IB}}^{\beta}$ for $\beta > 1$.

The effect of optimizing $\mathcal{L}_{\text{sq-IB}}^{\beta}$ is illustrated in Fig. 3, which shows two different IB curves (not strictly concave and strictly concave), isolines of $\mathcal{L}_{\text{sq-IB}}^{\beta}$ for different $\beta$, and points on the curves that maximize $\mathcal{L}_{\text{sq-IB}}^{\beta}$. For both curves, the maximizers for different $\beta$ are at different points of the curve.

---

[5]For simplicity, we consider only the differentiable points on the IB curve (a more thorough analysis would look at the superderivatives of $F$). Since $F$ is concave, however, it must be differentiable almost everywhere (Rockafellar, 2015, Thm 25.5).

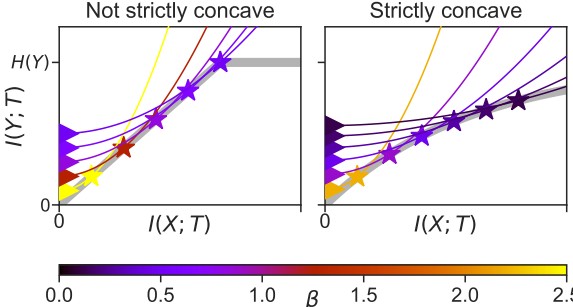

Figure 3: **Success of squared-IB functional**. Thick gray lines shows two possible IB curves, thin lines are isolines of $\mathcal{L}^{\beta}_{\text{sq-IB}}$ for different $\beta$ (colors indicate different $\beta$), stars indicate achievable $\langle I(X;T), I(Y;T) \rangle$ that maximize $\mathcal{L}^{\beta}_{\text{sq-IB}}$. For both the not strictly concave IB curve and the strictly concave curve, different $\beta$ values recover different solutions.

Finally, note that the IB curve is the Pareto front of the multi-objective optimization problem, $\{\max I(Y;T), \min I(X;T)\}$. $\mathcal{L}^{\beta}_{\text{sq-IB}}$ is one modification of $\mathcal{L}^{\beta}_{\text{IB}}$ that allows us to explore a non-strictly-concave Pareto front. However, it is not the only possible modification, and other alternatives may be considered. For a full treatment of multi-objective optimization, see Miettinen (1999).

## 5 ISSUE 2: ALL POINTS ON IB CURVE HAVE "UNINTERESTING" SOLUTIONS

It is often implicitly or explicitly assumed that IB optimal variables provide "useful" or "interesting" representations of the relevant information in $X$ about $Y$ (see also discussion and proposed criteria in (Amjad & Geiger, 2018)). While concepts like usefulness and interestingness are subjective, the intuition can be illustrated using the following example. Consider the ImageNet task (Deng et al., 2009), which labels images into 1000 classes, such as border collie, golden retriever, coffeepot, teapot, etc. It is natural to expect that as one explores the IB curve for the ImageNet dataset, one will identify useful compressed representations of the space of images. Such useful representations might, for instance, specify hard-clusterings that merge together inputs belonging to perceptually similar classes, such as border collie and golden retriever (but not border collie and teapot). Here we show that such intuitions will not generally hold when $Y$ is a deterministic function of $X$.

Recall the analysis in Section 3, where Eq. (6) defines the manifold of bottleneck variables $T_{\alpha}$ parameterized by $\alpha \in [0, 1]$. The manifold of $T_{\alpha}$, which spans the entire increasing portion of the IB curve, represents a mixture of two trivial solutions: a constant mapping to 0, and an exact copy of $Y$. Although the bottleneck variables on this manifold are IB-optimal, they do not offer interesting or useful representations of the input data. The compression offered by $T_{\alpha}$ arises by "forgetting" the input with some probability $1 - \alpha$, rather than performing any kind of useful hard-clustering, while the prediction comes from full knowledge of the function mapping inputs to outputs, $f : \mathcal{X} \to \mathcal{Y}$.

To summarize, when $Y$ is a deterministic function of $X$, the fact that a variable $T$ is on the IB curve does not necessarily imply that $T$ is an interesting compressed representation of $X$. At the same time, we do not claim that Eq. (6) provides unique solutions. There may also be IB-optimal variables that *do* compress $X$ in interesting and useful ways (however such notions may be formalized). Nonetheless, when $Y$ is a deterministic function of $X$, for any "interesting" $T$, there will also be an "uninteresting" $T_{\alpha}$ that achieves the same compression $I(X;T)$ and prediction $I(Y;T)$ values, and IB does not distinguish between the two. Therefore, identifying useful compressed representations must generally require the use of quality functions other than just IB.

## 6 ISSUE 3: NO TRADE-OFF AMONG DIFFERENT NEURAL NETWORK LAYERS

So far we've analyzed IB in terms of a single intermediate representation $T$ (e.g., a single hidden layer in a neural network). Recent research in machine learning, however, has focused on "deep" neural networks, with multiple successive hidden layers. What is the relationship between the compression and prediction achieved by these different layers? Recently, Shwartz-Ziv & Tishby (2017) suggested that due to SGD dynamics, different layers will explore a strict trade-off between compression and prediction: early layers will sacrifice compression (high $I(X;T)$) for good prediction (high $I(Y;T)$), while latter layers will sacrifice prediction (low $I(Y;T)$) for good compression (low $I(X;T)$). The authors demonstrated a strict trade-off using an artificial classification dataset,

in which $Y$ was defined to be a noisy function of $X$ (their Eq. 10 and Fig. 6). As we show, however, this outcome cannot generally hold when $Y$ is a deterministic function of $X$.

Recall that we consider IB in terms of the empirical distribution of inputs, hidden layers, and outputs given the training data (Section 2). Suppose that a classifier achieves 0 probability of error (i.e., classifies every input correctly) on training data. This event, which can only occur when $Y$ is a deterministic function of $X$, is in fact commonly observed in real-world deep neural networks (Zhang et al., 2017). In such cases, we show that while latter layers may have better compression than earlier layers, they cannot have worse prediction than earlier layers. Therefore, different layers can only demonstrate a *weak* trade-off between compression and prediction. (The same argument holds if the neural network achieves 0 probability of error on held-out testing data, as long as the information-theoretic measures are evaluated on the same held-out data distribution.)

Consider a neural network with $k$ hidden layers, where each successive layer is a (possibly stochastic) function of the preceding one, and let $T_1, T_2, \ldots, T_k$ indicate the activity of layer $1, 2, \ldots, k$. Given the predicted distribution of outputs, $q_\theta(\hat{Y} = \hat{y}|T_k = t_k)$, one can make a "point prediction" $\tilde{Y}$, e.g., by choosing the class with the highest predicted probability, $\tilde{Y} := \arg\max_{\hat{y}} q_\theta(\hat{y}|T_k)$. Given that the true output $Y$ is a function of $X$, the above architecture obeys the Markov condition

$$Y - X - T_1 - T_2 - \cdots - T_k - \hat{Y} - \tilde{Y}. \tag{9}$$

By the DPI, for any $i < j$ we have the inequalities

$$I(Y; T_i) \geq I(Y; T_j), \tag{10} \qquad\qquad I(X; T_i) \geq I(X; T_j). \tag{11}$$

Applying Fano's inequality (Cover & Thomas, 2012, p. 39) to the chain $Y - T_k - \tilde{Y}$ gives

$$\mathcal{H}(P_e) + P_e \log(|\mathcal{Y}| - 1) \geq H(Y|T_k), \tag{12}$$

where $|\mathcal{Y}|$ is the number of output classes, $P_e = \Pr(Y \neq \tilde{Y})$ is the probability of error, and $\mathcal{H}(x) := -x \log x - (1-x) \log(1-x)$ is the binary entropy function. Given our assumption that $P_e = 0$, Eq. (12) implies that $H(Y|T_k) = 0$ and thus $I(T_k; Y) = H(Y) - H(Y|T_k) = H(Y)$. Combining with Eq. (10) and $I(T_i; Y) \leq H(Y)$ means that $I(T_i; Y) = H(Y)$ for all $i = 1, \ldots, k$.

The above argument neither proves nor disproves the proposal in (Shwartz-Ziv & Tishby, 2017) that SGD dynamics favor hidden layer mappings that provide compressed representations of the input. Instead, our point is that for classifiers that achieve 0 prediction error, a strict compression/prediction trade-off is not possible. Even so, by Eq. (11) it is possible that latter layers have more compression than earlier layers while achieving the same prediction level (i.e., a weak trade-off). In terms of Fig. 1, this means that different layers will be on the flat part of the IB curve.

Note that some neural network architectures do not have a simple layer structure, in which each layer's activity is a stochastic function of the previous layer's, but rather allow information to flow in parallel across different channels (Szegedy et al., 2015) or to skip across some layers (He et al., 2016). The analysis in this section can still apply to such cases, as long as the different $T_1, \ldots, T_k$ are chosen to correspond to groups of neurons that satisfy the Markov condition in Eq. (9).

# 7 A REAL-WORLD EXAMPLE: MNIST

We demonstrate the three caveats using the MNIST dataset of hand-written digits. To do so, we use the "nonlinear IB" method (Kolchinsky et al., 2017), which uses gradient-descent-based training to minimize cross-entropy loss plus a differentiable non-parametric estimate of $I(X; T)$ (Kolchinsky & Tracey, 2017). We use this technique to maximize a lower bound on the IB Lagrangian, as explained in (Kolchinsky et al., 2017), as well as a lower bound on the squared-IB functional.

In our architecture, the bottleneck variable, $T \in \mathbb{R}^2$, corresponds to the activity of a hidden layer with two hidden units. Using a two-dimensional bottleneck variable facilitates easy visual analysis of its activity. The map from input $X$ to variable $T$ has the form $T = a_\theta(X) + Z$, where $Z \sim \mathcal{N}(0, \mathbf{I})$ is noise, and $a_\theta$ is a deterministic function implemented using three fully-connected layers: two layers with 800 ReLU units each, and a third layer with two linear units. Note that the stochasticity in the mapping from $X$ to $T$ makes our mutual information term, $I(X; T)$, well-defined and finite. The decoding map, $q_\theta(\hat{y}|t)$, uses a fully-connected layer with 800 ReLU units, followed by an

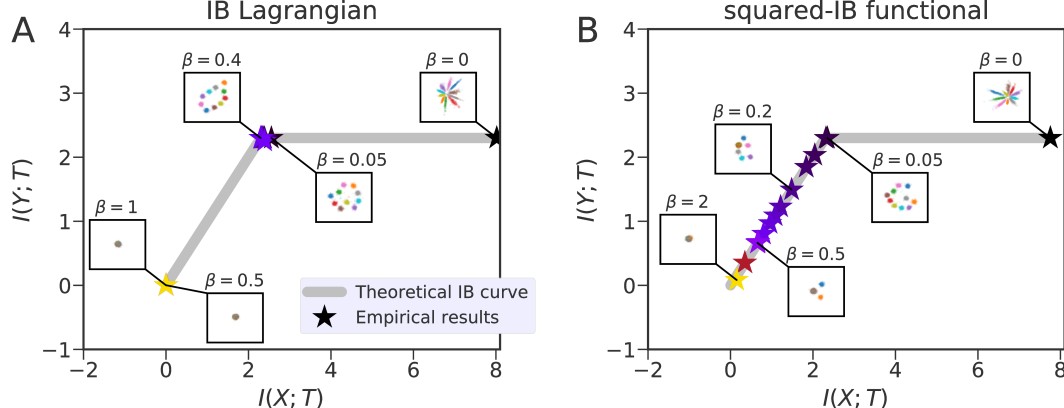

Figure 4: Theoretical and empirical IB curve found by maximizing the IB Lagrangian (A) and the squared-IB functional (B). Stars represent solutions recovered for different values of $\beta$. Insets show the bottleneck variable states (i.e., activity of the hidden layer) for different solutions, where colors represent inputs corresponding to different classes (i.e., different digits). MI values in nats.

output layer of 10 softmax units. Results are reported for training data. See Appendix A for more details and figures, including results on testing data. TensorFlow code can be found at `https://github.com/artemyk/ibcurve`.

Like other practical general-purpose IB methods, "nonlinear IB" is not guaranteed to find globally-optimal bottleneck variables due to factors like: (a) difficulty of optimizing the non-convex IB objective; (b) error in estimating $I(X;T)$; (c) limited model class of $T$ (i.e., $T$ must be expressible in the form of $T = a_\theta(X) + Z$); (d) mismatch between actual decoder $q_\theta(\hat{Y}|T)$ and optimal decoder $\bar{p}_\theta(Y|T)$ (see Section 2); and (e) stochasticity of training due to SGD. Nonetheless, in practice, the solutions discovered by nonlinear IB were very close to IB-optimal, and are sufficient to demonstrate the three caveats discussed in the previous sections.

**Issue 1: IB Curve cannot be explored using the IB Lagrangian**

We first demonstrate that the IB curve cannot be explored by maximizing the IB Lagrangian, but can be explored by maximizing the squared-IB functional, thus supporting the arguments in Section 4. Fig. 4 shows the theoretical IB curve for the MNIST dataset, as well as the IB curve empirically recovered by maximizing these two functionals (see also Fig. A5 for more details).

By optimizing the IB Lagrangian (Fig. 4A), we are only able to find three IB-optimal points:
(1) Maximum prediction accuracy and no compression, $I(Y;T) = H(Y)$, $I(X;T) \approx 8$ nats;
(2) Maximum compression possible at maximum prediction, $I(Y;T) = I(X;T) = H(Y)$;
(3) Total compression and zero prediction, $I(X;T) = I(Y;T) = 0$.

Note that the identified solutions are all very close to the theoretically-predicted IB-curve. However, the switch between the 2nd regime (maximal compression possible at maximum prediction) and 3rd regime (total compression and zero prediction) in practice happens at $\beta \approx 0.45$. This is different from the theoretical prediction, which states that this switch should occur at $\beta = 1.0$. The deviation from the theoretical prediction likely arises due to various practical details of our optimization procedure, as mentioned above. The switch from the 1st regime (no compression) to the 2nd regime happened as soon as $\beta > 0$, as predicted theoretically.

In contrast to the IB Lagrangian, by optimizing the squared-IB functional (Fig. 4B), we discover solutions located along different points on the IB curve for different values of $\beta$.

Additional insight is provided by visualizing the bottleneck variable $T \in \mathbb{R}^2$ (i.e., hidden layer activity) for both experiments. This is shown for different $\beta$ in the scatter plot insets in Fig. 4. As expected, the IB Lagrangian experiments displayed three types of bottleneck variables: non-compressed variables (regime 1), compressed variables where each of the 10 classes is represented by its own compact cluster (regime 2), and a trivial solution where all activity is collapsed to a single cluster (regime 3). For the squared-IB functional, a different behavior was observed: As $\beta$ increases, multiple classes become clustered together and the total number of clusters decreased.

Thus, nonlinear IB with the squared-IB functional learned to group $X$ into a varying number of clusters, in this way exploring the full trade-off between compression and prediction.

**Issue 2: All points on IB curve have "uninteresting" solutions**

By maximizing the squared-IB functional, we could find (nearly) IB-optimal solutions along different points of the IB curve. Here we show that such solutions do not provide particularly useful representations of the input data, supporting the arguments made in Section 5.

Note that "stochastic mixture"-type solutions, in particular the family $T_\alpha$ (Eq. (6)), are not in our model class, since they cannot be expressed in the form $T = a_\theta(X) + Z$. Instead, our implementation favors "hard-clusterings" in which all inputs belonging to a given output class are mapped to a compact, well-separated cluster in the activation space of the hidden layer (note that inputs belonging to multiple output classes may be mapped to a single cluster). For instance, Fig. 4B, shows solutions with 10 clusters ($\beta = 0.05$), 6 clusters ($\beta = 0.2$), 3 clusters ($\beta = 0.5$), and 1 cluster ($\beta = 2.0$). Interestingly, such hard-clusterings are characteristic of optimal solutions for deterministic IB (dIB), as discussed in Appendix B. At the same time, in our results, the classes are not clustered in any particularly meaningful or useful way, and clusters contain different combinations of classes for different solutions. For instance, the solution shown for squared-IB functional with $\beta = 0.5$ has 3 clusters, one of which contains the classes $\{0, 2, 3, 4, 5, 6, 8, 9\}$, another contains the class 1, and the last contains the class 7. However, in other runs for the same $\beta$ value, different clusterings of the classes arose. Moreover, because the different classes appear with close-to-uniform frequency in the MNIST dataset, any solution that groups the 10 classes into 3 clusters of size $\{8, 1, 1\}$ will achieve similar values of $I(X;T), I(Y;T)$ as the solution shown for $\beta = 0.5$.

**Issue 3: No trade-off among different neural network layers**

For both types of experiments, runs with $\beta = 0$ minimize cross-entropy loss only, without any regularization term that favors compression. (Such runs are examples of "vanilla" supervised learning, though with stochasticity in the mapping between the input and the 2-node hidden layer.) These runs achieved nearly-perfect prediction, so $I(Y;T) \approx H(Y)$. However, as shown in the scatter plots in Fig. 4, these hidden layer activations fell into spread-out clusters, rather than point-like clusters seen for $\beta > 0$. This shows that hidden layer activity was not compressed, in that it retained information about $X$ that was irrelevant for predicting the class $Y$, and fell onto the flat part of the IB curve.

Recall that our neural network architecture has three hidden layers before $T$, and one hidden layer after it. Due to the DPI inequalities Eqs. (10) and (11), the earlier hidden layers must have less compression than $T$, while the latter hidden layer must have more compression than $T$. At the same time, $\beta = 0$ runs achieve nearly 0 probability of error on the training dataset (results not shown), meaning that all layers must achieve $I(Y;T) \approx H(Y)$, the maximum possible. Thus, for $\beta = 0$ runs, as in regular supervised learning, the activity of the all hidden layers is located on the flat part of the IB curve, demonstrating a lack of a strict trade-off between prediction and compression.

## 8 CONCLUSION

The information bottleneck principle has attracted a great deal of attention in various fields, including information theory, cognitive science, and machine learning, particularly in the context of classification using neural networks. In this work, we showed that in any scenario where $Y$ is a deterministic function of $X$ — which includes many classification problems — IB demonstrates behavior that is qualitatively different from when the mapping from $X$ to $Y$ is stochastic. In particular, in such cases: (1) the IB curve cannot be recovered by maximizing the IB Lagrangian $I(Y;T) - \beta I(X;T)$ while varying $\beta$; (2) all points on the IB curve contain "uninteresting" representations of inputs; (3) multi-layer classifiers that achieve zero probability of error cannot have a strict trade-off between prediction and compression among successive layers, contrary to a recent proposal.

Our results should not be taken to mean that the application of IB to supervised learning is without merit. First, they do not apply to various non-deterministic classification problems where the output is stochastic. Second, even for deterministic scenarios, one may still wish to control the amount of compression during training, e.g., to improve generalization or robustness to adversarial inputs. In this case, however, our work shows that to achieve varying rates of compression, one should use a different objective function than the IB Lagrangian.

ACKNOWLEDGMENTS

We would like to thank the Santa Fe Institute for helping to support this research. Artemy Kolchinsky was supported by Grant No. FQXi-RFP-1622 from the FQXi foundation and Grant No. CHE-1648973 from the US National Science Foundation. Brendan D. Tracey was supported by the AFOSR MURI on multi-information sources of multi-physics systems under Award Number FA9550-15-1-0038.

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

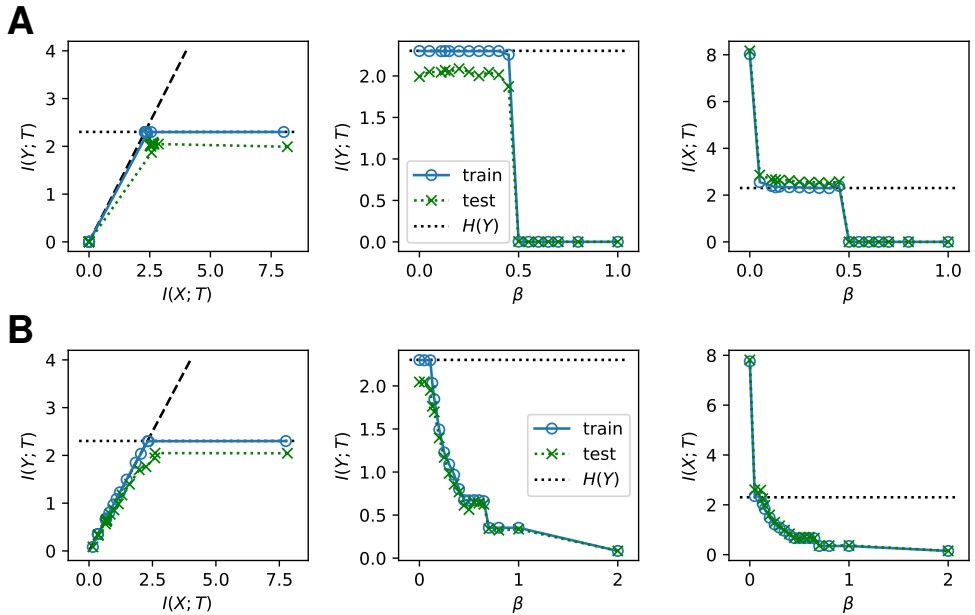

Figure A5: The top row (A) shows results for the IB Lagrangian, $I(Y;T) - \beta I(X;T)$, and the bottom row (B) shows results for the squared-IB functional, $I(Y;T) - \beta I(X;T)^2$. In each row, the left column shows the information plane (compression $I(X;T)$ versus prediction $I(Y;T)$), with the black dashed line showing the DPI bound $I(Y;T) \leq I(X;T)$; the middle column shows prediction $I(Y;T)$ as a function of $\beta$; the right column shows compression $I(X;T)$ as a function of $\beta$. In all plots, the solid blue line indicates values calculated for the training data set, the dashed green line indicates values calculated for the held-out testing data set, and the black dotted line indicates $H(Y) = \ln 10$. All information-theoretic quantities are plotted in nats.

## A   DETAILS OF MNIST EXPERIMENTS

In Section 7, we demonstrate our results on the MNIST dataset (LeCun et al., 1998). This dataset contains a training set of 60,000 images and a test set of 10,000 images, each labeled according to digit. $X \in \mathbb{R}^{784}$ is defined to be a vector of pixels for a single $28 \times 28$ image, and $Y \in \{0, 1, ..., 9\}$ is defined to be the class label. Our experiments were carried out using the "non-linear IB" method (Kolchinsky et al., 2017). $I(X;T)$ was computed using the kernel-based mutual information upper bound (Kolchinsky & Tracey, 2017; Kolchinsky et al., 2017) and $I(Y;T)$ was computed using the lower bound $I(Y;T) \geq H(Y) - \mathbb{E}_{\bar{p}_\theta(Y,T)}\left[ - \log q_\theta(\hat{Y}|T) \right]$ (see Eq. (5)).

The neural network was trained using the Adam algorithm (Kingma & Ba, 2014) with a mini-batch size of 128 and a learning rate of $10^{-4}$. Unlike the implementation in (Kolchinsky et al., 2017), the same mini-batch was used to estimate the gradients of both $I_\theta(X;T)$ and the cross-entropy term. Training was run for 200 epochs. At the beginning of each epoch, the order of training examples was randomized. To eliminate the effect of the local minima, for each possible value of $\beta$, we carried out 20 runs and then selected the run that achieved the best value of the objective function. TensorFlow code is available at `https://github.com/artemyk/ibcurve`.

Results for the MNIST dataset are shown in Fig. 4 and Fig. A5, computed for a range of $\beta$ values. Fig. A5 shows results for both training and testing datasets, though the main text focuses exclusively on training data. It can be seen that while the solutions found by IB Lagrangian jump discontinuously from the "fully clustered" solution ($I(X;T) = I(Y;T) = H(Y)$) to the trivial solution ($I(X;T) = I(Y;T) = 0$), solutions found by the squared-IB functional explore the trade-off in a continuous manner. See figure captions for details.

## B  DETERMINISTIC INFORMATION BOTTLENECK

Here we show that our analysis also applies to a recently-proposed (Strouse & Schwab, 2017) variant of IB called *deterministic IB (dIB)*. dIB replaces the standard IB compression cost, $I(X;T)$, with the entropy of the bottleneck variable, $H(T)$. This can be interpreted as operationalizing compression costs via a source-coding, rather than a channel-coding, scenario.

Formally, in dIB one is given random variables $X$ and $Y$. One then identifies bottleneck variables $T$ that obey the Markov condition $Y - X - T$ and maximize the *dIB Lagrangian*,

$$\mathcal{L}_{\text{dIB}}^{\beta}(T) := I(Y;T) - \beta H(T) \tag{A13}$$

for $\beta \in [0, 1]$, which can be considered as a relaxation of the constrained optimization problem

$$F_{\text{dIB}}(r) := \max_{T \in \Delta} I(Y;T) \quad \text{s.t.} \quad H(T) \le r \,. \tag{A14}$$

To guarantee that the compression cost is well-defined, $T$ is typically assumed to be discrete-valued (Strouse & Schwab, 2017). We call $F_{\text{dIB}}$ the *dIB curve*.

Before proceeding, we note that the inequality constraint in the definition of $F_{\text{dIB}}(r)$ can be replaced by an equality constraint,

$$F_{\text{dIB}}(r) := \max_{T \in \Delta} I(Y;T) \quad \text{s.t.} \quad H(T) = r \tag{A15}$$

We do so by showing that $F_{\text{dIB}}(r)$ is monotonically increasing in $r$. Consider any $T$ which maximizes $I(Y;T)$ subject to the constraint $H(T) = r$, and obeys the Markov condition $Y - X - T$. Now imagine some random variable $D$ which obeys the Markov condition $Y - X - T - D$, and define a new bottleneck variable $T' := (T, D)$ (i.e., the joint outcome of $T$ and $D$). We have

$$I(Y;T') = I(Y;T, D) = I(Y;T) + I(Y;D|T) = I(Y;T) \,,$$

where we've used the chain rule for mutual information. At the same time, $D$ can always be chosen so that $H(T') = H(T, D) = r'$ for any $r' \ge r$. Thus, we have shown that there are always random variables $T'$ that achieve at least $I(Y;T') = \max_{T:H(T)=r} I(Y;T)$ and have $H(T') > r$, meaning that $F_{\text{dIB}}(r)$ is monotonically increasing in $r$. This means the inequality constraint in Eq. (A14) can be replaced with an equality constraint.

As we will see, unlike the standard IB curve, $F_{\text{dIB}}$ is not necessarily concave. Since $F_{\text{dIB}}$ can be defined using equality constraints, one can rewrite maximization of $\mathcal{L}_{\text{dIB}}^{\beta}$ as $\max_T I(Y;T) - \beta H(T) = \max_r F_{\text{dIB}}(r) - \beta r$, the Legendre-Fenchel transform of $-F_{\text{dIB}}(r)$. By properties of the Legendre-Fenchel transform, the optimizers of $\mathcal{L}_{\text{dIB}}^{\beta}$ must lie on the concave envelope of $F_{\text{dIB}}$, which we indicate as $F_{\text{dIB}}^*$.

### B.1  THE dIB CURVE WHEN $Y$ IS A DETERMINISTIC FUNCTION OF $X$

As in standard IB, for a discrete-valued $Y$ we have the inequality

$$I(Y;T) \le H(Y) \,. \tag{A16}$$

However, instead of the standard IB inequality $I(Y;T) \le I(X;T)$, we now employ

$$I(Y;T) \le H(T) \,, \tag{A17}$$

which makes use of the assumption that $T$ is discrete-valued. The dIB curve will have the same bounds as those shown for the standard IB curve (Fig. 1), except that $H(T)$ replaces $I(X;T)$ on the horizontal axis.

Now consider the case where $Y$ is a deterministic function of $X$, i.e., $Y = f(X)$. It is easy to check that $T_{\text{copy}} := f(X) = Y$ achieves equality for both Eqs. (A16) and (A17), and thus lies on the dIB curve. Since $F_{\text{dIB}}$ is monotonically increasing, the dIB curve is flat and achieves $I(Y;T) = H(Y)$ for all $H(T) \ge H(Y)$.

We now consider the increasing part of the curve, $H(T) \in [0, H(Y)]$. We call any $T$ which is a deterministic function of $Y$ (that is, any $T = g(Y) = g(f(X))$, where $g$ is any deterministic function) a "hard-clustering" of $Y$. Any hard-clustering of $Y$ has $H(T) \le H(Y)$. At the same

time, any hard-clustering of $Y$ has $H(T|Y) = 0$, thus $I(Y;T) = H(T) - H(T|Y) = H(T)$, achieving the bound of Eq. (A17). Thus, any hard-clustering of $Y$ lies on the increasing part of the dIB curve. Note that any hard-clustering of $Y$ will also be a deterministic function of $X$, thus have $H(T|X) = 0$ and $I(X;T) = H(T) - H(T|X) = H(T) = I(Y;T)$, and will therefore also fall on the increasing part of the standard IB curve.

At the same time, the dIB curve cannot be composed entirely of hard-clustering, since — under the assumption that $Y$ is discrete-valued — there can only be a countable number of $T$'s that are hard-clusterings of $Y$. Thus, the dIB curve must also contain bottleneck variables that are not deterministic functions of $Y$, thus have $H(T|Y) > 0$ and $I(Y;T) < H(T)$, and do not achieve the bound of Eq. (A17). Geometrically-speaking, when $Y$ is a deterministic function of $X$, the dIB curve must have a "step-like" structure over $H(T) \in [0, H(Y)]$, rather than increasing smoothly like the standard IB curve. These results are shown schematically in Fig. A6, where blue dots indicate hard clusters of $Y$.

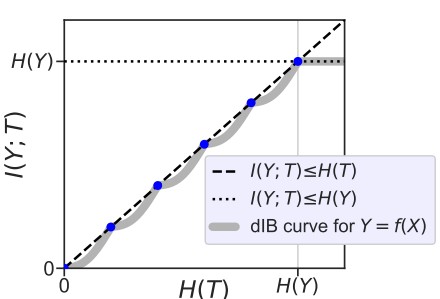

Figure A6: A schematic of the dIB curve. Dashed line is the bound $I(Y;T) \leq H(T)$, dotted line is $I(Y;T) \leq H(Y)$. When $Y$ is a deterministic function of $X$, the dIB curve saturate the second bound always. Furthermore, it also saturates the first bound for any $T$ that is a deterministic function of $Y$. The qualitative shape of the resulting dIB curve is shown as thick gray line.

As mentioned, optimizers of $\mathcal{L}_{\mathrm{dIB}}^{\beta}$ must lie on the concave envelope of $F_{\mathrm{dIB}}$, indicated by $F_{\mathrm{dIB}}^{*}$. Clearly, the step-like dIB curve that occurs when $Y$ is a deterministic function of $X$ is not concave, and only hard-clusterings of $Y$ lie on its concave envelope for $H(T) \in [0, H(Y)]$. In the analysis below, we will generally concern ourselves with optimizers which are hard-clusterings.

## B.2 THE THREE CAVEATS

We now briefly consider the three issues discussed in the main text, in the context of dIB. As usual, we assume that $Y$ is a deterministic function of $X$.

**Issue 1: dIB Curve cannot be explored using the dIB Lagrangian**

In analogy to the analysis done in Section 4, we use the inequalities Eqs. (A16) and (A17) to bound the dIB Lagrangian as

$$\mathcal{L}_{\mathrm{dIB}}^{\beta}(T) = I(Y;T) - \beta H(T) \leq (1-\beta)I(Y;T) \leq (1-\beta)H(Y).$$

Now consider the bottleneck variable $T_{\mathrm{copy}}$, for which $\mathcal{L}_{\mathrm{dIB}}^{\beta}(T_{\mathrm{copy}}) = (1-\beta)H(Y)$. Therefore, $T_{\mathrm{copy}}$ (or any one-to-one transformation of $T_{\mathrm{copy}}$) will maximize $\mathcal{L}_{\mathrm{dIB}}^{\beta}$ for all $\beta \in [0, 1]$. It is also straightforward to show that when $\beta = 0$, all bottleneck variables residing on the flat part of the dIB curve will simultaneously optimize the dIB Lagrangian. Similarly, one can show that all hard-clusterings of $Y$, which achieve the bound of Eq. (A17), will simultaneously optimize the dIB Lagrangian for $\beta = 1$. As before, this means that there is no one-to-one map between points on the dIB curve and optimizers of the dIB Lagrangian for different $\beta$.

As in Section 4, we propose to resolve this problem by maximizing an alternative objective function, which we call the *squared-dIB functional*,

$$\mathcal{L}_{\mathrm{sq\text{-}dIB}}^{\beta}(T) := I(Y;T) - \beta H(T)^{2}. \tag{A18}$$

We first demonstrate that any optimizer of the dIB Lagrangian must also be an optimizer of the squared-dIB functional. Consider that maximization of $\mathcal{L}_{\mathrm{dIB}}^{\beta}$ can be written as $\max_{T} I(Y;T) - \beta H(T) = \max_{r} F_{\mathrm{dIB}}^{*}(r) - r$. Then, for the point $\langle r, F_{\mathrm{dIB}}^{*}(r) \rangle$ on the dIB curve to maximize $\mathcal{L}_{\mathrm{dIB}}^{\beta}$, it must have $\beta \in \partial_{r} F_{\mathrm{dIB}}^{*}(r)$, where $\partial_{r}$ indicates the superderivative with regard to $r$. At the same time, for the point $\langle r, F_{\mathrm{dIB}}^{*}(r) \rangle$ to maximize $\mathcal{L}_{\mathrm{sq\text{-}dIB}}^{\beta}$, it must satisfy $0 \in \partial_{r} \left[ F_{\mathrm{dIB}}^{*}(r) - \beta r^{2} \right]$, or after rearranging,

$$2r\beta \in \partial_{r} F_{\mathrm{dIB}}^{*}(r). \tag{A19}$$

It is easy to see that if $\beta \in \partial_{r} F_{\mathrm{dIB}}^{*}(r)$ is satisfied, then Eq. (A19) is also satisfied under the transformation $\beta \mapsto \frac{\beta}{2r}$. Thus, any optimizer of $\mathcal{L}_{\mathrm{dIB}}^{\beta}$ will also optimize $\mathcal{L}_{\mathrm{sq\text{-}dIB}}^{\beta}$, given $\beta \mapsto \frac{\beta}{2r}$.

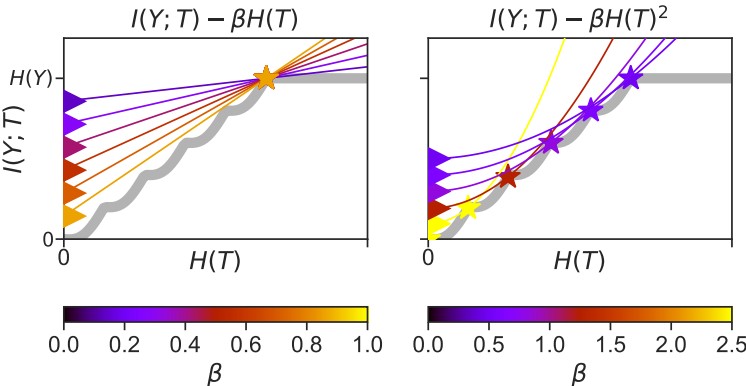

Figure A7: **Success of squared-dIB functional**. Colored lines indicate manifolds which have the same value of the dIB Lagrangian (left) and the squared-dIB functional (right) for different values of $\beta$, the stars indicate achievable $\langle H(T), I(Y;T)\rangle$ that maximize each function. For the dIB curve when $Y = f(X)$, different $\beta$ values recover different solutions only for the squared-dIB functional.

We now show that different hard-clusterings of $Y$ will optimize the squared-dIB functional for different values of $\beta$, meaning that we can explore the envelope of the dIB curve by optimizing $\mathcal{L}^{\beta}_{\text{sq-dIB}}$ while varying $\beta$. Formally, we show that for any given $\beta > 0$, the point

$$\langle H(T), I(Y;T)\rangle = \left\langle \frac{1}{2\beta}, \frac{1}{2\beta} \right\rangle \tag{A20}$$

on the dIB curve will be a unique maximizer of $\mathcal{L}^{\beta}_{\text{sq-dIB}}$ for the corresponding $\beta$. Consider the value of $\mathcal{L}^{\beta}_{\text{sq-dIB}}$ for any $T$ satisfying Eq. (A20),

$$\mathcal{L}^{\beta}_{\text{sq-dIB}}(T) = \frac{1}{2\beta} - \beta \left( \frac{1}{2\beta} \right)^2 = \frac{1}{2\beta} - \frac{1}{4\beta} = \frac{1}{4\beta}$$

Now consider the value of of $\mathcal{L}^{\beta}_{\text{sq-dIB}}$ for any other $T'$ on the dIB curve which has $H(T') \neq \frac{1}{2\beta}$,

$$\mathcal{L}^{\beta}_{\text{sq-dIB}}(T') = I(Y;T') - \beta H(T')^2$$

$$= I(Y;T') - \beta \left( \left( H(T') - \frac{1}{2\beta} \right)^2 + \frac{H(T')}{\beta} - \frac{1}{4\beta^2} \right)$$

$$\overset{(a)}{<} (I(Y;T') - H(T')) + \frac{1}{4\beta}$$

$$\overset{(b)}{\leq} \frac{1}{4\beta} = \mathcal{L}^{\beta}_{\text{sq-dIB}}(T).$$

Inequality $(a)$ comes from the assumption that $H(T') \neq \frac{1}{2\beta}$, thus $(H(T') - 1/(2\beta))^2 > 0$, while inequality $(b)$ comes from the bound $I(Y;T') \leq H(T')$. We have shown that $\mathcal{L}^{\beta}_{\text{sq-dIB}}(T') < \mathcal{L}^{\beta}_{\text{sq-dIB}}(T)$, meaning that any point on the dIB curve satisfying Eq. (A20) for a given $\beta$, assuming it exists, will be the unique maximizer of $\mathcal{L}^{\beta}_{\text{sq-dIB}}$. The situation is diagrammed visually in Fig. A7.

**Issue 2: All points on dIB curve have "uninteresting" solutions**

The family of bottleneck variables $T_\alpha$ defined in Eq. (6), i.e., the mixture of two trivial solutions, are no longer optimal from the point of view of dIB. However, as mentioned above, any $T$ that is a hard-clustering of $Y$ achieves the bound of Eq. (A17), and is thus on the dIB curve.

However, given a hard-clustering of $Y$, there is no reason for the clusters to obey any intuitions about semantic or perceptual similarity between grouped-together classes. To use the ImageNet example from Section 5, there is no reason for dIB to prefer a coarse-graining with "natural" groups like {border collie, golden retriever} and {teapot, coffeepot}, rather than a coarse-graining with groups like {border collie, teapot} and {golden retriever, coffeepot}, assuming those classes are of the same

size. Distinguishing between such different clusterings requires some similarity or distortion measure between classes, which is not provided by the standard information theoretic measures.

**Issue 3: No trade-off among different neural network layers**

In Section 6, we showed that for a neural network with many hidden layers and zero probability of error, the activity of the different layers will lie along the flat part of the standard IB curve, where there is no strict trade-off between compression $I(X;T)$ and prediction $I(Y;T)$. Note that any bottleneck variable that lies along the flat part of a standard IB curve will have $I(X;T) \geq H(Y)$ and $I(Y;T) = H(Y)$. Using the standard information-theoretic inequality $H(T) \geq I(X;T)$, the same bottleneck variable must therefore have $H(T) \geq H(Y)$ and $I(Y;T) = H(Y)$, thus also lying on the flat part of the dIB curve. Thus, for a neural network with many hidden layers and zero probability of error, the activity of the different layers will also lie along the flat part of the dIB curve, and not demonstrate any strict trade-off between compression $H(T)$ and prediction $I(Y;T)$.

## C   CAVEATS WHEN $Y$ IS APPROXIMATELY A DETERMINISTIC FUNCTION OF $X$

In this Appendix, we show that when $Y$ is a small perturbation away from being a deterministic function of $X$, the three caveats discussed in the main text persist in an approximate manner.

To derive our results, we first prove several useful theorems. In our proofs, we make central use of Thm. 17.3.3 from Cover & Thomas (2012), which states that, for two distributions $a$ and $b$ over the same finite set of outcomes $\mathcal{Y}$, if the $\ell_1$ distance is bounded as $|a - b|_1 \leq \epsilon \leq \frac{1}{2}$, then $|H(a) - H(b)| \leq -\epsilon \log \frac{\epsilon}{|\mathcal{Y}|}$. We will also use the weaker bound $|H(a) - H(b)| \leq \log |\mathcal{Y}|$, which is based on the maximum and minimum entropy for any distribution over outcomes $\mathcal{Y}$.

**Theorem 1.** *Let $Z$ be a random variable (continuous or discrete), and $Y$ a random variable with a finite set of outcomes $\mathcal{Y}$. Consider two joint distributions over $Z$ and $Y$, $p_{ZY}$ and $\tilde{p}_{ZY}$, which have the same marginal over $Z$, $p(z) = \tilde{p}(z)$, and obey $|p_{ZY} - \tilde{p}_{ZY}|_1 \leq \epsilon \leq \frac{1}{2}$. Then,*

$$|H(p(Y|Z)) - H(\tilde{p}(Y|Z))| \leq -\epsilon \log \frac{\epsilon}{|\mathcal{Y}|^3} .$$

*Proof.* For this proof, we will assume that $Z$ is a continuous-valued. In case it is discrete-valued, the below proof applies after replacing all integrals over the outcomes of $Z$ with summations.

Let $\hat{\epsilon} := |p_{ZY} - \tilde{p}_{ZY}|_1 = \int \sum_y |p_{ZY}(z,y) - \tilde{p}_{ZY}(z,y)| \, dz$ be the actual $\ell_1$ distance between $p_{ZY}$ and $\tilde{p}_{ZY}$, and note that $\hat{\epsilon} \leq \epsilon$ by assumption. Without loss of generality, we assume that $\hat{\epsilon} > 0$. Then, define the following probability distribution,

$$q(z) := \frac{1}{\hat{\epsilon}} p(z) \left| p_{Y|Z=z} - \tilde{p}_{Y|Z=z} \right| = \frac{1}{\hat{\epsilon}} p(z) \sum_y |p(y|z) - \tilde{p}(y|z)| .$$

Note that $q(z)$ integrates to 1 and is absolutely continuous with respect to $p(z)$ ($p(z) = 0$ implies $q(z) = 0$), and that $\left| p_{Y|Z=z} - \tilde{p}_{Y|Z=z} \right| = \hat{\epsilon} \frac{q(z)}{p(z)}$. When then have

$$|H(p(Y|z)) - H(\tilde{p}(Y|z))| \leq \begin{cases} -\hat{\epsilon} \frac{q(z)}{p(z)} \log \frac{\hat{\epsilon}}{|\mathcal{Y}|} \frac{q(z)}{p(z)} & \text{if } \hat{\epsilon} \frac{q(z)}{p(z)} \leq \frac{1}{2} \\ \log |\mathcal{Y}| & \text{otherwise} \end{cases} ,$$

where the first case follows from (Cover & Thomas, 2012, Thm. 17.3.3), and the second case by considering minimum vs. maximum possible entropy values.

We now bound the difference in conditional entropies, using $[\cdot]$ to indicate the Iverson bracket,

$$H(p(Y|Z)) - H(\tilde{p}(Y|Z)) = \int p(z) \left[ H(p(Y|z)) - H(\tilde{p}(Y|z)) \right] \, dz \leq$$
$$\int p(z) \left[ \hat{\epsilon} \frac{q(z)}{p(z)} \leq \frac{1}{2} \right] \left( -\hat{\epsilon} \frac{q(z)}{p(z)} \log \frac{\hat{\epsilon}}{|\mathcal{Y}|} \frac{q(z)}{p(z)} \right) dz + \log |\mathcal{Y}| \int p(z) \left[ \hat{\epsilon} \frac{q(z)}{p(z)} > \frac{1}{2} \right] \, dz . \quad \text{(A21)}$$

We upper bound the first integral in Eq. (A21) as

$$\int p(z) \left[ \hat{\epsilon} \frac{q(z)}{p(z)} \leq \frac{1}{2} \right] \left( -\hat{\epsilon} \frac{q(z)}{p(z)} \log \frac{\hat{\epsilon}}{|\mathcal{Y}|} \frac{q(z)}{p(z)} \right) \, dz$$

$$\leq \int p(z) \left( -\hat{\epsilon} \frac{q(z)}{p(z)} \log \frac{\hat{\epsilon}}{|\mathcal{Y}|} \frac{q(z)}{p(z)} \right) dz \tag{A22}$$

$$= -\hat{\epsilon} \int q(z) \left( \log \frac{\hat{\epsilon}}{|\mathcal{Y}|} + \log \frac{q(z)}{p(z)} \right) dz$$

$$= -\hat{\epsilon} \log \frac{\hat{\epsilon}}{|\mathcal{Y}|} - \hat{\epsilon} D_{\mathrm{KL}}(q(Z) \| p(Z)) \leq -\hat{\epsilon} \log \frac{\hat{\epsilon}}{|\mathcal{Y}|} \tag{A23}$$

where in Eq. (A22) we dropped the Iverson bracket, in Eq. (A23) we used the definition of KL divergence, and then its non-negativity.

To upper bound the second integral in Eq. (A21), observe that

$$\int p(z) \left[ \hat{\epsilon} \frac{q(z)}{p(z)} > \frac{1}{2} \right] \, dz = \int p(z) \left[ \left| p_{Y|Z=z} - \tilde{p}_{Y|Z=z} \right| > \frac{1}{2} \right] \, dx$$

$$\leq \int p(z) \, 2 \left| p_{Y|Z=z} - \tilde{p}_{Y|Z=z} \right| \left[ \left| p_{Y|Z=z} - \tilde{p}_{Y|Z=z} \right| > \frac{1}{2} \right] \, dz$$

$$\leq 2 \int p(z) \left| p_{Y|Z=z} - \tilde{p}_{Y|Z=z} \right| \, dz \leq 2\hat{\epsilon} \tag{A24}$$

Combining Eqs. (A21), (A23) and (A24) gives

$$H(p(Y|Z)) - H(\tilde{p}(Y|Z)) \leq -\hat{\epsilon} \log \frac{\hat{\epsilon}}{|\mathcal{Y}|} + 2\hat{\epsilon} \log |\mathcal{Y}| = -\hat{\epsilon} \log \frac{\hat{\epsilon}}{|\mathcal{Y}|^3}$$

The theorem follows by noting that $-\hat{\epsilon} \log \hat{\epsilon}$ is monotonically increasing for $\hat{\epsilon} \leq \frac{1}{2}$, and by assumption $\hat{\epsilon} \leq \epsilon \leq \frac{1}{2}$. $\qquad \square$

In the following theorems, we use the notation $I_q(Z; Y)$ to indicate mutual information between $Z$ and $Y$ evaluated under some joint distribution $q$ over $Z$ and $Y$.

**Theorem 2.** *Let $Z$ be a random variable (continuous or discrete), and $Y$ a random variable with a finite set of outcomes. Consider two joint distribution over $X \times Y$, $p_{XY}$ and $\tilde{p}_{XY}$, which have the same marginal over $Z$, $p(z) = \tilde{p}(z)$, and obey $|p_{ZY} - \tilde{p}_{ZY}|_1 \leq \epsilon \leq \frac{1}{2}$. Then,*

$$|I_p(Z; Y) - I_{\tilde{p}}(Z; Y)| \leq -2\epsilon \log \frac{\epsilon}{|\mathcal{Y}|^2} .$$

*Proof.* We first bound the $\ell_1$ distance between $p_Y$ and $\tilde{p}_Y$,

$$|p_Y - \tilde{p}_Y|_1 = \sum_y |p_Y(y) - \tilde{p}_Y(y)| = \sum_y \left| \int \left( p_{ZY}(z, y) - \tilde{p}_{XY}(z, y) \right) \, dz \right|$$

$$\leq \sum_y \int |p_{ZY}(z, y) - \tilde{p}_{ZY}(z, y)| \, dz = |p_{ZY} - \tilde{p}_{ZY}|_1 \leq \epsilon$$

By (Cover & Thomas, 2012, Thm. 17.3.3),

$$|H(p(Y)) - H(\tilde{p}(Y))| \leq -\epsilon \log \frac{\epsilon}{|\mathcal{Y}|} . \tag{A25}$$

We now bound the magnitude of the difference of mutual informations as

$$|I_p(Z; Y) - I_{\tilde{p}}(Z; Y)| = |H(p(Y)) - H(\tilde{p}(Y)) - (H(p(Y|Z)) - H(\tilde{p}(Y|Z)))|$$

$$\leq |H(p(Y)) - H(\tilde{p}(Y))| + |H(p(Y|Z)) - H(\tilde{p}(Y|Z))|$$

$$\leq -\epsilon \log \frac{\epsilon}{|\mathcal{Y}|} - \epsilon \log \frac{\epsilon}{|\mathcal{Y}|^3} = -2\epsilon \log \frac{\epsilon}{|\mathcal{Y}|^2} ,$$

where in the last line we've used Eq. (A25) and Theorem 1. $\qquad \square$

**Theorem 3.** *Let $X$ be a random variable (continuous or discrete), and $Y$ a random variable with a finite set of outcomes $\mathcal{Y}$. Consider two joint distribution over $X$ and $Y$, $p_{XY}$ and $\tilde{p}_{XY}$, which have the same marginal over $X$, $p(x) = \tilde{p}(x)$ and obey $|p_{XY} - \tilde{p}_{XY}|_1 \leq \epsilon \leq \frac{1}{2}$. Let $F(r)$ and $\tilde{F}(r)$ indicate the IB curves for $p_{XY}$ and $\tilde{p}_{XY}$ respectively, as defined in Eq. (1). Then for any $r > 0$,*

$$|F(r) - \tilde{F}(r)| \leq -2\epsilon \log \frac{\epsilon}{|\mathcal{Y}|^2} \, .$$

*Proof.* For any compression level $r > 0$, let $T$ and $\tilde{T}$ be optimal bottleneck variables for $p_{XY}$ and $\tilde{p}_{XY}$ respectively,

$$I(X;T) \leq r \quad ; \quad I_p(Y;T) = F(r)$$
$$I(X;\tilde{T}) \leq r \quad ; \quad I_{\tilde{p}}(Y;\tilde{T}) = \tilde{F}(r) \, .$$

Let $T$ be defined by the stochastic map $q(t|x)$, and note that $p_T(t) = \int p(x)q(t|x) \, dx = \int \tilde{p}(x)q(t|x) \, dx = \tilde{p}_T(t)$. Consider two joint distributions over $T$ and $Y$,

$$p_{TY}(t, y) = \int p_{XY}(x, y)q(t|x) \, dx \quad ; \quad \tilde{p}_{TY}(t, y) = \int \tilde{p}_{XY}(x, y)q(t|x) \, dx \, ,$$

and observe that

$$|p_{TY} - \tilde{p}_{TY}|_1 = \int dt \sum_y \left| \int p_{XY}(x, y)q(t|x) \, dx - \int \tilde{p}_{XY}(x, y)q(t|x) \, dx \right|$$

$$\leq \int dt \sum_y \int q(t|x) \, |p_{XY}(x, y) - \tilde{p}_{XY}(x, y)| \, dx$$

$$\leq \sum_y \int |p_{XY}(x, y) - \tilde{p}_{XY}(x, y)| \, dx \leq \epsilon$$

We now apply Theorem 2 with $Z = T$ and $Y = Y$ to give $I_p(Y;T) + 2\epsilon \log \frac{\epsilon}{|\mathcal{Y}|^2} \leq I_{\tilde{p}}(Y;T)$. At the same time, by definition of the IB curve, it must be that $I_{\tilde{p}}(Y;T) \leq \tilde{F}(r) = I_{\tilde{p}}(Y;\tilde{T})$. Combining the last two inequalities gives

$$I_p(Y;T) + 2\epsilon \log \frac{\epsilon}{|\mathcal{Y}|^2} \leq I_{\tilde{p}}(Y;\tilde{T}) \, . \tag{A26}$$

The same argument can be repeated while exchanging the roles of $T$ and $\tilde{T}$ (thus defining the joint distributions $p_{\tilde{T}Y}$ and $\tilde{p}_{\tilde{T}Y}$, etc.). This gives the inequality

$$I_{\tilde{p}}(Y;\tilde{T}) + 2\epsilon \log \frac{\epsilon}{|\mathcal{Y}|^2} \leq I_p(Y;T) \, . \tag{A27}$$

The theorem follows by combining Eqs. (A26) and (A27), and using the relations $I_p(Y;T) = F(r)$, $I_{\tilde{p}}(Y;T) = \tilde{F}(r)$. $\qquad\square$

It is important to emphasize that having a small $\ell_1$ distance between the joint distributions $p_{XY}$ and $p_{XY}$ does not imply that the conditional distributions $p_{Y|X=x}$ and $\tilde{p}_{Y|X=x}$ have to be close for all $x$. In fact, the conditional distributions can be arbitrarily different for some $x$, as long as such $x$ do not have much probability under $p(x)$.

We use these theorems to demonstrate how, if the joint distribution of $X$ and $Y$ is $\epsilon$-close to having $Y$ be a deterministic function of $X$, then the three caveats discussed in the main text all apply in an approximate manner, meaning that can only be avoided up to order $\mathcal{O}(-\epsilon \log \epsilon)$.

## C.1 THE THREE CAVEATS WHEN $Y$ IS APPROXIMATELY A DETERMINISTIC FUNCTION OF $X$

**Issue 1: IB curve cannot be explored using the IB Lagrangian**

In the main text, we demonstrated that when $Y$ is a deterministic function of $X$, the single point $\langle H(Y), H(Y) \rangle$ on the information plane optimizes the IB Lagrangian for all $\beta \in [0, 1]$. We now consider the case where the joint distribution of $X$ and $Y$ is $\epsilon$-close to having $Y$ be a deterministic function of $X$. The next theorem shows that in this case, optimizers of the IB Lagrangian must still be close to $\langle H(Y), H(Y) \rangle$. Formally, for any fixed $\beta \in (0, 1)$, the maximum possible distance from $\langle H(Y), H(Y) \rangle$ scales as $\mathcal{O}(-\epsilon \log \epsilon)$, though the scaling constant increases as $\beta$ approaches 0 or 1. This implies that for small $\epsilon$, it will difficult to find solutions substantially different from $\langle H(Y), H(Y) \rangle$, as one would have to search using $\beta$ very close to 0 or very close to 1.

**Theorem 4.** *Let $X$ be a random variable (continuous or discrete), and $Y$ a random variable with a finite set of outcomes $\mathcal{Y}$. Let $\tilde{p}_{XY}$ be a joint distribution over $X$ and $Y$ under which $Y = f(X)$. Let $p_{XY}$ be a joint distribution over $X$ and $Y$ which has the same marginal over $X$ as $\tilde{p}_{XY}$, $p(x) = \tilde{p}(x)$, and obeys $|p_{XY} - \tilde{p}_{XY}|_1 \leq \epsilon \leq \frac{1}{2}$.*

*Then, for any $\beta \in (0, 1)$ and $T$ which maximizes the $p_{XY}$ IB Lagrangian $I_p(Y; T) - \beta I_p(X; T)$,*

$$-\frac{\gamma}{1 - \beta} \leq I_p(X; T) - H(p(Y)) \leq \frac{\gamma}{\beta} \tag{A28}$$

$$-\frac{\gamma}{1 - \beta} \leq I_p(Y; T) - H(p(Y)) \leq 0 \tag{A29}$$

*where $\gamma := -3\epsilon \log \epsilon + 5\epsilon \log |\mathcal{Y}|$.*

*Proof.* First, note that if $|p_{XY} - \tilde{p}_{XY}|_1 \leq \epsilon$, then $|p_Y - \tilde{p}_Y|_1 \leq \epsilon$ and (see proof of Theorem 2)

$$|H(p(Y)) - H(\tilde{p}(Y))| \leq -\epsilon \log \frac{\epsilon}{|\mathcal{Y}|}. \tag{A30}$$

Let $F$ and $\tilde{F}$ indicate the IB curves for $p_{XY}$ and $\tilde{p}_{XY}$, respectively, defined as in Eq. (1). Because $\tilde{p}_{XY}$ is deterministic, the IB curve $\tilde{F}$ is the piecewise linear function $\tilde{F}(x) = \min\{x, H(\tilde{p}(Y))\}$. Given Eq. (A30), this means that

$$\tilde{F}(H(p(Y))) \geq H(p(Y)) + \epsilon \log \frac{\epsilon}{|\mathcal{Y}|}. \tag{A31}$$

We now to bound $F(H(p(Y)))$ as

$$\begin{aligned}
F(H(p(Y))) &\geq \tilde{F}(H(p(Y))) + 2\epsilon \log \frac{\epsilon}{|\mathcal{Y}|^2} \\
&\geq H(p(Y)) + 2\epsilon \log \frac{\epsilon}{|\mathcal{Y}|^2} + \epsilon \log \frac{\epsilon}{|\mathcal{Y}|} \\
&= H(p(Y)) - \gamma,
\end{aligned} \tag{A32}$$

where in the first line we've used Theorem 3, and in the second line we used Eq. (A31).

Now consider $T$, the bottleneck variable that optimizes the $p_{XY}$ IB Lagrangian for some $\beta \in (0, 1)$. By concavity of the IB curve $F$, all points on the IB curve must fall below the line with slope $\beta$ that passes through the point $\langle I_p(X; T), I_p(Y; T) \rangle$ on the information plane, including the point $\langle H(p(Y)), F(H(p(Y))) \rangle$. Formally, we write this condition as

$$I_p(Y; T) + \beta \left( H(p(Y)) - I_p(X; T) \right) \geq F(H(p(Y))) \geq H(p(Y)) - \gamma, \tag{A33}$$

where the second inequality uses Eq. (A32). Note that by the DPI, $I_p(X; T) \geq I_p(Y; T)$. Combining with Eq. (A33) gives

$$\begin{aligned}
I_p(X; T) + \beta \left( H(p(Y)) - I_p(X; T) \right) &\geq H(p(Y)) - \gamma \\
(\beta - 1) \left( H(p(Y)) - I_p(X; T) \right) &\geq -\gamma \\
H(p(Y)) - I_p(X; T) &\leq \frac{\gamma}{1 - \beta}.
\end{aligned} \tag{A34}$$

At the same time, by properties of mutual information, $H(p(Y)) \geq I_p(Y;T)$. Combining again with Eq. (A33) gives

$$H(p(Y)) + \beta \left( H(p(Y)) - I_p(X;T) \right) \geq H(p(Y)) - \gamma$$

$$H(p(Y)) - I_p(X;T) \geq -\frac{\gamma}{\beta} \tag{A35}$$

The result Eq. (A28) follows immediately from Eq. (A34) and Eq. (A35) and rearranging.

We now derive Eq. (A29). The upper bound arises from basic properties of mutual information, $H(p(Y)) \geq I_p(Y;T)$. To derive the lower bound, we rewrite Eq. (A33) as

$$I_p(Y;T) \geq H(p(Y)) - \gamma + \beta(I_p(X;T) - H(p(Y))) \tag{A36}$$

$$\geq H(p(Y)) - \gamma - \frac{\beta}{1-\beta}\gamma = H(p(Y)) - \frac{\gamma}{1-\beta}, \tag{A37}$$

where in the second line we used Eq. (A28) and then combined terms. $\qquad\square$

**Issue 2: All points on the IB curve are $\mathcal{O}(-\epsilon \log \epsilon)$ away from trivial solutions.**

In the main text, we show that when $Y$ is a deterministic function of $X$, there are trivial bottleneck variables which are optimal for all points of the IB curve. We now consider the case where the joint distribution of $X$ and $Y$ is $\epsilon$-close to having $Y$ be a deterministic function of $X$. In this case, by using Theorem 3, it is straightforward to show that there are trivial bottleneck variables that are $\mathcal{O}(-\epsilon \log \epsilon)$ away from being optimal along all points on the IB curve.

**Theorem 5.** *Let $X$ be a random variable (continuous or discrete), and $Y$ a random variable with a finite set of outcomes $\mathcal{Y}$. Let $\tilde{p}_{XY}$ be a joint distribution over $X$ and $Y$ under which $Y = f(X)$. Let $p_{XY}$ be a joint distribution over $X$ and $Y$ which has the same marginal over $X$ as $\tilde{p}_{XY}$, $p(x) = \tilde{p}(x)$, and obeys $|p_{XY} - \tilde{p}_{XY}|_1 \leq \epsilon \leq \frac{1}{2}$.*

*Then, for any $r > 0$, there is some value of $\alpha \in [0,1]$ such that the bottleneck variable $T_\alpha$ defined by the conditional distribution $q_\alpha(T_\alpha = t | X = x) = \alpha\delta(t, f(x)) + (1-\alpha)\delta(t,0)$ satisfies*

$$I_p(T_\alpha;X) \leq r \quad ; \quad I_p(T_\alpha;Y) \geq F(r) - \gamma,$$

*where $\gamma := -3\epsilon\log\epsilon + 5\epsilon\log|\mathcal{Y}|$.*

*Proof.* First, note that if $|p_{XY} - \tilde{p}_{XY}|_1 \leq \epsilon$, then $|p_Y - \tilde{p}_Y|_1 \leq \epsilon$ and

$$|H(p(Y)) - H(\tilde{p}(Y))| \leq -\epsilon\log\frac{\epsilon}{|\mathcal{Y}|}. \tag{A38}$$

(See proof of Theorem 2.)

Let $F(r)$ and $\tilde{F}(r)$ indicate the IB curve for $p_{XY}$ and $\tilde{p}_{XY}$, respectively, as defined in Eq. (1). By properties of mutual information and Eq. (A38), we have

$$F(r) \leq H(p(Y)) \leq H(\tilde{p}(Y)) - \epsilon\log\frac{\epsilon}{|\mathcal{Y}|}. \tag{A39}$$

We now consider two cases.

The first case is when $r \leq H(\tilde{p}(Y))$. Recall that because $\tilde{p}_{XY}$ is deterministic, the IB curve $\tilde{F}$ is the piecewise linear function $\tilde{F}(x) = \min\{x, H(\tilde{p}(Y))\}$. Thus, we choose an $\alpha$ so that $I_p(X;T_\alpha) = r$ and, using Theorem 3,

$$I_p(X;T_\alpha) = \tilde{F}(r) \geq F(r) + 2\epsilon\log\frac{\epsilon}{|\mathcal{Y}|^2} \geq F(r) - \gamma.$$

Second, when $r > H(\tilde{p}(Y))$, we choose $\alpha = 1$ and note that $I_p(T_\alpha;X) = H(\tilde{p}(Y)) \leq r$. It can be verified (see for instance proof of Theorem 3) that the joint distributions $p_{T_\alpha Y}(t,y) = \int p_{XY}(x,y)q_\alpha(t|x) \ dx$ and $\tilde{p}_{T_\alpha Y}(t,y) = \int \tilde{p}_{XY}(x,y)q_\alpha(t|x) \ dx$ satisfy the conditions of Theorem 2 with $Z = T_\alpha, Y = Y$. Applying that theorem gives

$$I_p(Y;T_\alpha) \geq I_{\tilde{p}}(Y;T_\alpha) + 2\epsilon\log\frac{\epsilon}{|\mathcal{Y}|^2} = H(\tilde{p}(Y)) + 2\epsilon\log\frac{\epsilon}{|\mathcal{Y}|^2} \geq H(p(Y)) - \gamma,$$

where the last inequality uses Eq. (A39). $\qquad\square$

**Issue 3: Prediction/compression trade-off can be most of order $\mathcal{O}(-\epsilon \log \epsilon)$**

In the main text, we consider a neural network consisting of $k$ layers, and show that when $Y$ is a deterministic function of $X$, there can be no strict prediction/compression trade-off between the different layers. We now consider the case where the joint distribution of $X$ and $Y$ is $\epsilon$-close to having $Y$ be a deterministic function of $X$.

As before, we let $X$ be a (continuous or discrete) random variable, and $Y$ a random variable with a finite set of outcomes $\mathcal{Y}$. Let $\tilde{p}_{XY}$ be a joint distribution over $X$ and $Y$ under which $Y = f(X)$. Let $p_{XY}$ be a joint distribution over $X$ and $Y$ which has the same marginal over $X$ as $\tilde{p}_{XY}$, $p(x) = \tilde{p}(x)$, and obeys $|p_{XY} - \tilde{p}_{XY}|_1 \leq \epsilon \leq \frac{1}{2}$.

We assume that for each input $X$, our classifier outputs the prediction $\tilde{Y} = f(X)$ — that is, it predicts according to the deterministic input-output mapping $f$. For small $\epsilon$, this is an optimal prediction strategy in terms of $P_e$, the probability of error. We write $P_e$ for this classifier as

$$P_e = \int p(x) \left( 1 - p_{Y|X=x}(f(x)|x) \right) dx$$

$$= \frac{1}{2} \int p(x) \left( \left( 1 - p_{Y|X=x}(f(x)|x) \right) + \sum_{y:y\neq f(x)} p_{Y|X=x}(y|x) \right)$$

$$= \frac{1}{2} \int \sum_y p(x) \left| p_{Y|X=x}(y|x) - \tilde{p}_{Y|X=x}(y|x) \right| dx$$

$$= \frac{1}{2} |p_{XY} - \tilde{p}_{XY}|_1 \leq \frac{1}{2}\epsilon$$

We now restate Fano's inequality (Eq. (12)) for the activity of the last layer, $T_k$, as

$$H(Y|T_k) \leq \mathcal{H}(P_e) + P_e \log(|\mathcal{Y}| - 1) \leq -2P_e \log \frac{2P_e}{|\mathcal{Y}|} = -\epsilon \log \frac{\epsilon}{|\mathcal{Y}|}, \tag{A40}$$

where the second inequality is found in Zhang (2007).

As before, we note that latter layers may compress input information more than earlier ones, in that $I(T_k; X)$ may be smaller than $I(T_1; X)$. Moreover, when $Y$ is not exactly a deterministic function of $X$, then in principle a strict prediction/compression trade-off between different layers is possible (i.e., $I(T_i; Y)$ can decrease for latter layers). However, if the joint distribution of $X$ and $Y$ is $\epsilon$-close to having $Y = f(X)$, then the magnitude this trade-off is limited in size,

$$I(T_1; Y) - I(T_k; Y) = H(Y|T_k) - H(Y|T_1) \leq -\epsilon \log \frac{\epsilon}{|\mathcal{Y}|},$$

where we use non-negativity of conditional entropy and Eq. (A40).

