# OpenReview forum: "Caveats for information bottleneck in deterministic scenarios"
_ICLR.cc/2019/Conference_

### Official Review · AnonReviewer2 · 2018-10-28
**The present work interestingly clarifies several counter-intuitive behaviors of the information bottleneck (IB) method for the learning of a deterministic rule. We note, however, that the necessity of noise for its application to supervised learning was already known.**

**Rating:** 6
**Confidence:** 4

**Review:**

This work analyses the information bottleneck (IB) method applied to the supervised learning of a deterministic rule Y=f(X).

The idea as I understood it is as follows:
1) In a first section the authors discuss the relationship between supervised learning through minimization of the empirical cross entropy and the maximization of the empirical mutual information with an intermediate latent variable T.
2) They show that in the case of a deterministic rule, the information bottleneck curve has a simple shape, piecewise linear, and is not strictly concave.
3) They show that the optimization of the IB Lagrangian for different \beta does not lead to a point by point exploration of the IB curve.
4) They propose a cure to the previous issue by introducing the squared IB Lagrangian.
5) They exhibit uninteresting representations (noisy versions of the output Y) that are on the IB curve.
6) They show that multiple successive representations (like in DNNs), have identical predicting power (mutual information with output Y) when they allow for perfect prediction.
7) They use the IB method to train a neural net on MNIST, using the Kolchinsky estimate of the mutual informations.
	- they show that the optimization of the squared IB reaches more different points on the IB curve,
	- but that these representations are possibly uninteresting (hard clustering of uneven numbers of grouped classes)
	- they show that for large enough value of beta, zero error is reached.

The necessity of noise in the IB theory has been already pointed out by (Gilad-Bachrach et al., 2003; Shwartz-Ziv et al.  2017), although the more thorough analysis proposed here is novel. In practice, besides a few recent propositions (Kolchinsky et al., 2017; Alemi et al., 2016; Chalk et al., 2016) the IB Lagrangian is not a usual objective function for supervised learning. The motivation and impact of this work studying deterministic rules is therefore not completely convincing.

Further pros and cons:

Pros:
- The discussion is generally well written.
- This work provides in depth clarification of the counter-intuitive behaviors of the IB method in the case to the learning of a deterministic rule.
- These are demonstrated with experiments conducted on the MNIST dataset for concreteness.

Cons:
- The fact that multiple successive representations have identical predicting power when the prediction error is zero, was already observed for example in Shwartz-Ziv et al.  2017. It is not clear why this should be considered as an issue. It also seems to be a straightforward observation when restricting to the empirical measure on the training set.
- The fact that the entire IB curve is not explored point by point by the IB Lagrangian is not necessarily an issue for learning. In the experiments of the present paper, the results seem to suggest that the interesting intermediate representations (separation in 10 compact clusters of the MNIST classes) is actually easier to obtain (large range of \beta) optimizing the IB Lagrangian rather than the proposed squared IB Lagrangian.

Questions:
- Do the authors know of an application where the full probing of the IB curve would be necessary?
- In Section 2, when injecting the decomposition of the prediction density q(y|x) over the intermediate variable t in eq (3) was a Jensen inequality replaced by an equality?

---

> ### Author Response · Authors · 2018-11-14
> **Revision and response (part 1)**
>
> We thank the reviewer for the thoughtful reading and comments. We address several of the reviewer’s criticisms point by point (broken into three comments for length):
>
> > The necessity of noise in the IB theory has been already pointed
> > out by (Gilad-Bachrach et al., 2003; Shwartz-Ziv et al.  2017),
> > although the more thorough analysis proposed here is novel.
>
> We agree with the reviewer that Gilad-Bachrach et al., 2003 and Shwartz-Ziv et al.  2017 also discussed the role of noise in the mapping from X to Y in IB. Gilad-Bachrach et al., 2003 showed that IB curve is not strictly concave without noise. However, we believe that this result may not be widely known, nor -- more importantly -- that it implies that the IB Lagrangian fails in deterministic scenarios was not appreciated. Shwartz-Ziv et al. discussed the fact that when Y=f(X), mutual information between input and output doesn’t reflect the “the complexity of the function f(x) or the class of functions it comes from” (where complexity could be understood in terms of, e.g., VC dimension) [section 2.4, “The crucial role of noise”]. This is an interesting issue, but is orthogonal to the issues discussed in our paper. However, we have added a sentence in the Introduction to draw attention Shwartz-Ziv et al.’s discussion of this other, interesting caveat.
>
> We note that even without noise, the IB curve still exists and is well-defined via the constrained optimization problem (Eq. 1 in our paper, Eq. 2 in Gilad-Bachrach et al.), or via the alternative objective function (squared-IB Lagrangian that we propose). However, it has not been previously recognized that in deterministic settings, this IB curve will be full of trivial solutions.
>
> Finally, we have seen various recent articles that apply IB concepts to supervised learning in which class labels are completely deterministic and which do not seem to be aware of any of the possible caveats mentioned in our article (or that of Gilad-Bachrach et al. or Shwartz-Ziv et al.).
>
>
> > In practice, besides a few recent propositions (Kolchinsky et al., 2017;
> > Alemi et al., 2016; Chalk et al., 2016) the IB Lagrangian is not a
> > usual objective function for supervised learning. The motivation
>  > and impact of this work studying deterministic rules is therefore
>  > not completely convincing.
>
> We see our paper as being about the fundamental properties of IB, in particular when applied to deterministic settings. Neural networks is one area where IB has recently been receiving a lot of attention, and where deterministic mappings are very common, and seems like a natural area of application.
>
> However, we believe the work can be of interest to a diverse community of researchers, including:
>
> (1) Those using IB in various applied settings. This includes not just recent work deep learning (where the IB Lagrangian has been suggested as an objective function), but numerous applications of IB in speech recognition (Yaman et al, 2012; Hecht et al, 2005), image recognition (Winn et al., 2005), video (Hsu et al., 2006), distributional clustering (Slonim et al, 2000), network coding (Zeitler et al., 2008), etc.
>
> (2) Those working in theoretical machine learning, for example by investigating the idea that stochastic gradient descent may “implicitly” optimize the IB Lagrangian (Shwartz-Ziv et al., Zhao 2018 [arXiv:1803.07980]), or analyzing properties of IB-optimal representations (Amjad et al. 2018, arXiv:1802.09766)
>
> (3) Those analyzing theoretical properties of IB in other fields, e.g., from the point of view of rate distortion (Harremoes et al., 2007), data compression (Cardinal, 2003), source doing (Courtade et al. 2011, arXiv:1106.0032), adaptive quantization (Lazebnik, 2009), etc.
>
> We recognize, however, that the title and tone of our article suggests that its its primary domain of application is supervised learning. We have tweaked the title and some of the article text to emphasize that we see supervised learning as one important application area of our results, among others.

---

> > ### Author Response · Authors · 2018-11-14
> > **Response (part 2)**
> >
> > > Cons:
> > > - The fact that multiple successive representations have identical
> > > predicting power when the prediction error is zero, was already
> > > observed for example in Shwartz-Ziv et al.  2017. It is not clear why
> > > this should be considered as an issue. It also seems to be a
> > > straightforward observation when restricting to the empirical measure
> > > on the training set.
> >
> > We are unable to find a statement in Shwartz-Ziv et al. that clearly says that multiple layers will have the same predictive power when prediction error of the whole network is 0. However, we agree with the reviewer that this could be inferred from some of the other results in that paper. We think it is useful to highlight this behavior clearly, because we believe it is not widely recognized in the community (e.g., some may expect their layers to explore a strict compression/prediction trade-off, e.g., as shown in Fig 6 of Shwartz-Ziv et al.). We also think it is interesting in that it demonstrates another instance of IB behaving in a qualitatively different way specifically when Y is a deterministic function of X.
> >
> > We agree with the reviewer that this behavior is not necessarily problematic in itself. We have changed the title and some of the text of the manuscript to refer to “Caveats” rather than “Pathologies” of IB, in large because of this third issue.
> >
> >
> > > - The fact that the entire IB curve is not explored point by point
> > > by the IB Lagrangian is not necessarily an issue for learning. In
> > > the experiments of the present paper, the results seem to suggest
> > >  that the interesting intermediate representations (separation in
> > > 10 compact clusters of the MNIST classes) is actually easier to
> > > obtain (large range of \beta) optimizing the IB Lagrangian rather
> > >  than the proposed squared IB Lagrangian.
> >
> > We appreciate this comment, and agree that if the goal is to simply find the “corner point” (i.e., maximal compression with no loss of prediction), then the IB Lagrangian works well and doesn’t require careful selection of beta. Moreover, we think this is a useful insight that emerges from the results and are paper, and we have added some text to Section 4 to highlight this (as a side note, this might suggests some potentially interesting algorithms for finding the corner point, e.g. setting beta initially to 0 and then slowly increasing during optimization of the IB Lagrangian...).
> >
> > However, in many cases (see next point), one’s goal is in fact to explore the IB curve, and there the IB Lagrangian fails.
> >
> > Moreover, we think it is commonly thought that the IB Lagrangian provides a general way to explore the IB curve, where by changing beta one changes the balance between compression and prediction. The fact that it sticks to a single corner point for deterministic Y is at the very least quite surprising.
> >
> >
> > > Questions:
> > > - Do the authors know of an application where the full probing
> > > of the IB curve would be necessary?
> >
> > We are familiar with several use cases in which one is interested in exploring the IB curve (note that we do not claim that the research mention below all involves deterministic scenarios, though some of it does, but rather only that it involves approaches that might be applied in such scenarios).
> >
> > In machine learning, Alemi et al. (2017) proposed that training with IB can provide robustness against adversarial inputs, and suggested that different points on the IB curve provide different levels of robustness; similarly, Alemi et al. (2018) proposed that IB can be used to detect out-of-distribution data. Finding a good trade-off between robustness/detectability and prediction accuracy, or at the very least evaluating such claims, requires exploration of the IB curve. There are also some connections between IB and generalization error (Shamir et al., 2010, Vera et al., 2018), so one may want to adaptively balance between training cross-entropy loss and generalization error guarantees.
> >
> > IB has been proposed as a method for distributional clustering, see e.g., Slonim et al., (2000) and cites thereof. In this case, exploring the IB curve allows one to adaptively control the resolution of the clustering.
> >
> > IB has an important interpretation from the point of view of rate-distortion/channel-coding (if Alice has access to X and a capacity-limited information channel to Bob, and Bob wants to optimally predict Y, then Bob does best by receiving an IB-optimal bottleneck variable). For this reason, IB has drawn attention from various researchers in coding theory, quantization, compression and rate distortion (Cardinal, 2003; Zeitler et al., 2008; Harremoes et al., 2007; Courtade et al. 2011, arXiv:1106.0032, Lazebnik, 2009, etc.) In such cases, it is centrally important to be able to explore the IB curve, since the optimal representations need to be adapted to available channel capacity (which can vary).

---

> > > ### Author Response · Authors · 2018-11-14
> > > **Response (part 3)**
> > >
> > > - In Section 2, when injecting the decomposition of the prediction
> > > > density q(y|x) over the intermediate variable t in eq (3) was a
> > > > Jensen inequality replaced by an equality?
> > >
> > > Here we exchanged integral/summation for an expectation over an appropriate distribution. So we did not move anything into the logarithm, and it is in fact an equality.
> > >
> > >
> > > In addition to the points above, we would like to draw the reviewer’s attention to a new paragraph and Appendix C (made partly in response to reviewer 3), where we prove that our results apply approximately when the relationship between X and Y is very close to, but not exactly, deterministic.  In particular, we show that if Y is epsilon-close to being deterministic, then the three caveats we discuss in the text can only be avoided by O(-epsilon log epsilon), in a formal sense defined in Appendix C. We believe this adds to the technical contribution of our manuscript.

---

> > > > ### Comment · AnonReviewer2 · 2018-11-23
> > > > **Answer satisfactory, except one remaining point.**
> > > >
> > > > I am in line with the authors change of tone/title of the article.
> > > > I appreciate their response pointing to cases where exploring the complete IB curve was investigated.
> > > > Also I note their taking into account of comments made on the interpretation of their results and highlighting that IB works well to find the corner point in Section 4.
> > > >
> > > > My comment on eq (3) was probably not clear, and that is likely why it is not answered in a satisfactory way. It is not a key point, but still I would appreciate a clarification. I explain better my issue: $q_\theta(y|x)$ is defined 3 lines above eq. (3). It is then plugged into the definition of ${\cal L}_{CE}(\theta)$ one line before eq. (3). But in order to get the first equality in eq. (3) one needs to change the log and the integral, isn't it? (I have not issue with the 2nd equality in eq. (3) .... which is what the authors misunderstood I think).

---

> > > > > ### Author Response · Authors · 2018-11-26
> > > > > **Response to remaining point**
> > > > >
> > > > > Actually, we did misunderstand the reviewer's comment about Eq. (3).  The reviewer is in fact correct, and there was a mistake in the manuscript.
> > > > >
> > > > > Cross-entropy loss is equal to $1/N \sum_i log q_theta(y_i | x_i)$ -- as we previously stated -- only when there is no stochasticity in the network.  In the general case of a stochastic neural network, cross-entropy loss is the expected log likelihood of the true output given the last layer's activity, or given the activity of any layer of which the last layer's activity is a deterministic function, and Eq. (3) holds under this condition.
> > > > >
> > > > > We have uploaded a revised manuscript with this correction. We thank the reviewer for their careful reading.

---

### Official Review · AnonReviewer1 · 2018-11-04
**A good paper on information bottleneck for machine learning**

**Rating:** 8
**Confidence:** 4

**Review:**

This paper is about issues that arise when applying Information Bottleneck (IB) concepts to machine learning, more precisely in deterministic supervised learning such as classification (deterministic in the sense that the target function to estimate is deterministic: it associates each example to one true label only, and not to a distribution over labels).
Namely:
(1) the "Information Bottleneck curve" cannot be computed with the Information Bottleneck Lagrangian approach (because of optimization landscape issues: optimization of such a piecewise-linear function with a linear penalty will always yield the same optimum whatever the slope of the penalty is [same story as L1 vs. L0]);
(2) there are many solutions to the optimization of the IB Lagrangian for any given compression/performance ratio (i.e. for any given beta in the IB Lagrangian method: I(Y,T)/I(X,T)) and some of them are provably trivial; thus optimizing just the IB Lagrangian does not imply that the solution will be interesting, and better (or complementary) criteria are needed.

Another point discussed also is about the successive layers of perfect classifiers (neural networks), in which I(Y,T) remains constant while I(X,T) decreases.


Pros:
- the paper is well written, mostly self-contained, and easy to read (for someone familiar with information theory);
- all mathematical points are detailed and well explained, with sufficient introduction;
- the writing is compact, the paper is dense, and given the page limit this is a good information/compression compromise;)
- information bottleneck is a topic of prime interest in the community these days;
- the two first problems described ((1) and (2)) are original, interesting contributions to the field, of particular interest for people interested in applying information bottleneck concepts to supervised learning;
- the solution brought to the IB Lagrangian issues is simplistic though efficient (squaring I(X,T) so that it's not linear in I(X,T) anymore).


Cons:
- not much.

Remarks:
- there exist recent papers tackling the information bottleneck concept for neural networks from a variational perspective, which enables them to compute exactly the mutual informations (such as "Compressing Neural Networks using the Variational Information Bottleneck" by Dai & al., ICML 2018); I have not seen these papers cited in the article, nor discussed (nor used); I feel it would be appropriate, either in the general literature section, either for discussing how to compute in practice the mutual informations (exact values vs. estimates or lower bounds as here).
- at first reading, I had found the tone of the beginning of the paper (first section) a bit aggressive, though this feeling disappeared later. Maybe rephrase some expressions that might be wrongly perceived?
- About multilabel classification (end of section 2): multilabel classification can still be seen as with deterministic expected outputs, if considered as a task from X to P(Y) (power set of Y, i.e. set of all possible subsets of labels).
- As in practice T is constrained to belong to a particular space of functions (neural network layer with predefined architecture): how does this impact the study? For instance the T_alpha in equation (5) are not reachable anymore; the optimization space for the IB Lagrangian is different; etc. Which properties/conclusions can be kept, and which ones cannot?
- What about sampling on the other part of the IB curve, the horizontal one (same I(Y,T) for various I(X,T))? Would it bring any insight, and how to do it?
- A side remark about applying IB to neural networks: What about neural networks that are not a "linear" chain of layers (i.e. most networks now)? i.e. Inception, ResNet, U-nets, etc., where computational flows are parallel, sometimes keeping full information till the end. For instance in a U-net, meant for image processing, features computed at the beginning at a full pixelic resolution are communicated to the last layer. This is not an image classification task though, as predictions are made for each pixel; still, given an input image X, there is only one correct output Y, so, still in the deterministic supervised classification problem.

---

> ### Author Response · Authors · 2018-11-14
> **Revision and response**
>
> We thank the reviewer for their supportive words and helpful suggestions. We address the reviewer’s remarks point by point (split into two responses for space).
>
> > - there exist recent papers tackling the information bottleneck concept for
> > neural networks from a variational perspective, which enables them to compute
> > exactly the mutual informations (such as "Compressing Neural Networks using
> > the Variational Information Bottleneck" by Dai & al., ICML 2018); I have not
> > seen these papers cited in the article, nor discussed (nor used); I feel it
> > would be appropriate, either in the general literature section, either for
> > discussing how to compute in practice the mutual informations (exact values
> > vs. estimates or lower bounds as here).
>
> We have added some text to section “2) Supervised Classification and IB”, where we highlight that our theoretical results are independent of how mutual information between neural network layers is estimated, but that this is an important and active area of research (including recent work by Belghazi et al., 2018; Goldfeld et al., 2018; Dai et al., 2018; Gabrié et al., 2018). For our empirical results, we use the estimator proposed in Kolchinsky et al., (2017).
>
>
> > - at first reading, I had found the tone of the beginning of the paper (first
> > section) a bit aggressive, though this feeling disappeared later. Maybe
> > rephrase some expressions that might be wrongly perceived?
>
> We appreciate this suggestion. We do not intend for our work to be viewed as an attack on IB. To soften the tone, we have rephrased several sentences in the abstract and introduction that may have been perceived as overly aggressive. Moreover, we have (partly in response to reviewer 2) replaced ‘pathology’ with ‘caveat’ throughout the text. We believe that this wording is more appropriate, particularly for the third issue (i.e., the lack of strict prediction/compression trade-off between different layers of a neural net), which is more of an ‘unexpected behaviour’ than it a ‘pathology’.
>
>
> > - About multilabel classification (end of section 2): multilabel classification can
> > still be seen as with deterministic expected outputs, if considered as a task from
> > X to P(Y) (power set of Y, i.e. set of all possible subsets of labels).
>
> We fully agree with the reviewer’s point, and have updated the manuscript accordingly.
>
>
> > - As in practice T is constrained to belong to a particular space of functions
> > (neural network layer with predefined architecture): how does this impact the
> > study? For instance the T_alpha in equation (5) are not reachable anymore; the
> > optimization space for the IB Lagrangian is different; etc. Which
> > properties/conclusions can be kept, and which ones cannot?
>
> These are great questions. An in-depth analysis of the effects of model constraints on T would be very interesting, especially for the types of constraints that characterize real-world architectures. At the same time, it is hard to say something meaningful for the general case, since different constraints on the set of T can produce arbitrarily different 'constrained IB-curves'. Due to the page limit, we must leave this topic for future work. We do note that in our experimental results, where we use a very standard MLP network architecture which does not include T_\alpha itself in the space of model, we witness all three issues discussed in the main text.
>
>
> > - What about sampling on the other part of the IB curve, the horizontal one
> > (same I(Y,T) for various I(X,T))? Would it bring any insight, and how to do
> > it?
>
> We expect it should be straightforward to design objective functions that encourage exploration of the flat part of the curve. However, points on the flat part of the curve are (weakly) Pareto dominated by the ‘corner point’. From the conceptual point of view of IB, in which it is assumed to always be better to have low I(X;T) all else being equal, we cannot think of why one might want to do this.

---

> > ### Author Response · Authors · 2018-11-14
> > **Response (part 2)**
> >
> > (cont'd)
> >
> > > - A side remark about applying IB to neural networks: What about neural
> > > networks that are not a "linear" chain of layers (i.e. most networks now)?
> > > i.e. Inception, ResNet, U-nets, etc., where computational flows are parallel,
> > > sometimes keeping full information till the end. For instance in a U-net,
> > > meant for image processing, features computed at the beginning at a full
> > > pixelic resolution are communicated to the last layer. This is not an image
> > > classification task though, as predictions are made for each pixel; still,
> > > given an input image X, there is only one correct output Y, so, still in the
> > > deterministic supervised classification problem.
> >
> > This is an important point.  In fact, our results can still apply to such cases, as long as T is chosen to be a set of neurons (or more generally internal state variables) that separate the input from the predicted output (in the conditional independence sense), so that one can write
> >     Pr( predicted_output | input_vector ) = Pr( predicted_output | t ) Pr( t | input_vector)
> > Thus, for example, T can be taken to be the set of all non-input-node neurons, or alternatively the set of output neurons themselves, which can be done in almost any neural architecture.
> >
> > Our analysis of caveat 3 (i.e., the lack of a strict trade-off between different neural network layers) does rely on some kind of “decomposability” of the network architecture, but doesn’t necessarily depend on a strictly layered architecture. As long as the different T_i are chosen so their corresponding neurons form a Markov chain from inputs to predicted outputs, the analysis can be applied (For example, a particular T_i could includes neurons that are part of several parallel streams in an Inception-type architecture.)
> >
> > We have added some text to the manuscript to highlight both of these points.
> >
> > In addition to the points above, we would like to draw attention the reviewer’s to a new paragraph and Appendix C (made partly in response to reviewer 3), where we prove that our results apply approximately when the relationship between X and Y is very close to, but not exactly, deterministic.  In particular, we show that if Y is epsilon-close to being deterministic, then the three caveats we discuss in the text can only be avoided by O(-epsilon log epsilon), in a formal sense defined in Appendix C.

---

> > > ### Comment · AnonReviewer1 · 2018-11-20
> > > **Discussion**
> > >
> > > Thank you for your detailed answers.
> > > About the literature: thanks for citing these additional, related papers. And sorry for mis-remembering the paper by Dai & al (it does not compute the exact mutual information as I thought; yet it is a great paper in other respects and still deserves to be cited here).
> > > About the horizontal part of the IB curve: your answer is convincing indeed.
> > > The additional study (Appendix C) completes nicely the paper. It seems that "Issue 1" disappears as such for approximately-only deterministic target functions (if one chooses suitable values of beta), which was expected somehow.

---

### Official Review · AnonReviewer3 · 2018-11-06
**Pathologies in information bottleneck for deterministic supervised learning**

**Rating:** 2
**Confidence:** 4

**Review:**

SUMMARY:
This paper is about potential problems of the information bottleneck principle in cases where the output variable Y is a deterministic function of the inputs X. Such a deterministic relationship between outputs and inputs induces the problem that the the IB "information curve" (i.e. I(T;Y) as a function of I(X;T)) is piece-wise linear and, thus, no longer strictly concave, which is crucial for non-degenerate ("interesting") solutions. The authors argue that most real classification problems indeed show such a deterministic relation between the class labels and the inputs X, and they explore several issues that result from such pathologies.

EVALUATION:
In my opinion, the whole story could be summarized as follows: if  Y is
a deterministic function of p-dimensional inputs X, then the joint distribution P(X,Y) is
degenerate in that its support lies in a space of dimension p (an not p+1 as it would be in the non-degenerate situation), and this is the source of all pathologies observed. As a consequence, only the cumulative distribution is defined, but there is no density with respect to the Lebesgue measure of R^{p+1}. Thus, one has to be careful when defining the mutual information I(X,Y), which explains the problems with the IB information curve (which should asymptotically converge to I(X;Y) as I(X;T) gets large. Another consequence of this degeneracy concerns the latent variable interpretation of the IB: if T is treated as a latent variable (as, for instance, in the "deep" IB models) then we have the conditional independence relation "Y independent of X given T", which simply makes no sense if Y is deterministic in X (there is, of course, a deeper underlying problem here: the IB problem is difficult in that it is difficult to define a geneative model with a faithful DAG...).
Analyzing situations in which Y = f(X) (with f being a deterministic function) is certainly interesting from a theoretic point of view, but I am not convinced that this analysis is truly relevant for practical problems.
In particular, I strongly disagree with the statement that "in most classification problems, the labels Y are a deterministic function of X". I would rather argue that the opposite is the case, because I don't think that there are too many such problems with zero Bayes error rate.  In particular, I would argue that digit recognition problems like MNIST so not have deterministic labels, since there will always be images of handwritten characters that will give room for interpretation...

---

> ### Author Response · Authors · 2018-11-14
> **Revision and response**
>
> We thank the reviewer for their comments. However, there appears to be some misunderstanding, which we attempt to address with our revision and comments below (response broken into 2 comments for space reasons).
>
> > EVALUATION: In my opinion, the whole story could be summarized as follows: if
> > Y is a deterministic function of p-dimensional inputs X, then the joint
> > distribution P(X,Y) is  degenerate in that its support lies in a space of
> > dimension p (an not p+1 as it would be in the non-degenerate situation), and
> > this is the source of all pathologies observed. As a consequence, only the
> > cumulative distribution is defined, but there is no density with respect to
> > the Lebesgue measure of R^{p+1}. Thus, one has to be careful when defining the
> > mutual information I(X,Y), which explains the problems with the IB information
> > curve (which should asymptotically converge to I(X;Y) as I(X;T) gets large.
>
> It is true that there has been some recent work (Saxe et al. 2018; Amjad et al., 2018) on the degeneracies that occur when T (the bottleneck variable, such as the hidden layer) is a continuous-valued and deterministic function of a continuous-valued input layer.
>
> However, the caveats described in our paper are unrelated to this problem, and arise even when all mutual information terms and probability distributions are well-defined and finite. In our case, Y (the output class) is a discrete random variable over a finite set (call this set [Y]) and the joint distribution of X and Y is a mixed continuous-discrete distribution over R^p \times [Y]. Moreover, the conditional distribution p(y|x) is a discrete probability distribution for every x. In this case, the mutual information is given by I(X;Y) = H(Y), and is bounded between 0 and log |Y|.
>
> Based on the reviewer’s comments, we have attempted to clarify our work by inserting text into the Introduction, which states that our caveats are not the result of degenerate distributions or poorly defined mutual information.
>
>
> > Another consequence of this degeneracy concerns the latent variable
> > interpretation of the IB: if T is treated as a latent variable (as, for
> > instance, in the "deep" IB models) then we have the conditional independence
> > relation "Y independent of X given T", which simply makes no sense if Y is
> > deterministic in X
>
> Unfortunately, we are not sure we understand the reviewer’s comment.
>
> For clarity, we emphasize that the usual Markov condition for IB is “Y is independent of T given X” (Y - X - T).  This remains true in a neural network with hidden layers, where the hidden layer T separates input layer X from *predicted outputs*, since X still separates T from the true output Y (we use Y to refer to the true output).
>
> We do show in our paper that when Y is deterministic in X, the IB curve will be populated by bottleneck variables on it that obey both Y - X - T (as all bottleneck variables must) and X - Y - T, such as our family T_alpha [see discussion around our Eq. 5].
>
> Finally, it is true that T_alpha for alpha=1 (in which case T_alpha is simply equal to Y) does also obey the independence condition “Y independent of X given T” (X - T - Y). This bottleneck variable sits at the “corner point” of the piecewise linear IB curve. However, we disagree with the reviewer that “ ‘Y independent of X given T’ ... makes no sense if Y is deterministic in X”.  If T=Y, as in this one particular case, then Y will in fact be conditionally independent of X given T, under the usual definition of conditional independence.
>
>
> > (there is, of course, a deeper underlying problem here: the
> > IB problem is difficult in that it is difficult to define a geneative model
> > with a faithful DAG...).
>
> Unfortunately, we are not sure we understand the reviewer’s point.  We will say, however, that in IB, one begins by assuming that X and Y are provided, then selects among T that obey T - X - Y. This fits naturally into the setting of supervised learning, where X represents the input, Y represents the true outputs, and T can refer to any intermediate representations (e.g., hidden layer neurons). The form of the mapping from X to the true output Y does not matter, nor does the form of the representation from X to T.  Any standard ML discriminative model will obey this Markov condition.

---

> > ### Author Response · Authors · 2018-11-14
> > **Response (part 2)**
> >
> >
> > > Analyzing situations in which Y = f(X) (with f being
> > > a deterministic function) is certainly interesting from a theoretic point of
> > > view, but I am not convinced that this analysis is truly relevant for
> > > practical problems.  In particular, I strongly disagree with the statement
> > > that "in most classification problems, the labels Y are a deterministic
> > > function of X". I would rather argue that the opposite is the case, because I
> > > don't think that there are too many such problems with zero Bayes error rate.
> > > In particular, I would argue that digit recognition problems like MNIST so not
> > > have deterministic labels, since there will always be images of handwritten
> > > characters that will give room for interpretation...
> >
> > In our paper, when considering the application of IB to classification problems, we have defined things in relation to the empirical distribution p(X,Y), as found in the training dataset and/or testing dataset, and it a fact that many classification datasets are deterministic. For example, neither the training nor testing dataset for MNIST contain more than one exemplar of an image, and each image is labelled with a single class only, and so the empirical distribution is deterministic. We believe that many supervised learning problems are commonly seen as deterministic, in which the challenge is to find the one true label for each input.
> >
> > At the same time, we recognize that (to use the MNIST example) there may be handwritten digits “out there in the world” that cannot be deterministically assigned to a single class due to subjective interpretations of images. Our arguments in the paper are motivated by thinking about the empirical distributions of train/test data because it is somewhat difficult to speculate  about such cases. Nonetheless, we appreciate the reviewers position, and we have softened some of the language, to emphasize that we believe many (rather than most) classification problems have a deterministic nature.  We have also added some new text to the Introduction and a new appendix (Appendix C) where we show that our results apply approximately when the relationship between X and Y is very close to, but not exactly, deterministic.  In particular, when Y is epsilon-close to being deterministic (in L1 norm of the joint X,Y distribution), then our caveats will still apply in the sense that: (1) it is hard to explore the IB curve by optimizing the IB Lagrangian, because all optimizers will fall within O(-epsilon log epsilon) of a single ‘corner' point on the information plane; (2) there are ‘uninteresting' trivial solutions which are no more than O(-epsilon log epsilon) away from being optimal along all points on the IB curve; (3) different layers of a neural networks can trade-off at most O(-epsilon log epsilon) amount of prediction. Note that our constraint on the L1 distance of the joint distribution from a deterministic mapping allows either for a small global non-determinism for all X, or for a small number of X to have a very non-deterministic relations (as in the hypothetical digits example).

---

### Public Comment · (anonymous) · 2018-11-07
**relevance?**

I am not sure I see the relevance of this paper: Nobody uses the the deterministic IB in deep learning, and the fact that there are issues in deterministic setting has already been argued by Saxe and co-workers at ICLR last year. Is this a straw-man?

---

> ### Author Response · Authors · 2018-11-14
> **Relevance**
>
> Thank you for your comment. We are not sure what the comment “nobody uses the deterministic IB” refers to --- deterministic IB typically refers to a specialized variant of IB proposed by Strouse and Schwab (arXiv:1604.00268), and it is not the main focus of our paper (though we discuss it in Appendix B).
>
> We believe there is a quite active interest in IB in the deep learning community, as evidenced by recent literature. Our results also apply outside of deep and/or supervised learning contexts, and we believe they hold interest for those working with IB in various other contexts (information theory, coding theory, etc., see response to Reviewer 2 below).
>
> The paper by Saxe et al. 2018, did not discuss inherent issues in IB itself, but rather evaluated the claim by Shwartz-Ziv that training of deep nets tends to ‘implicitly’ carry out IB, because of SGD dynamics. This is not the question considered in this paper. The “issues in deterministic setting” discussed in Saxe et al. concern that the fact that MI(input layer ; hidden layer) can be unbounded when the input->hidden layer map is deterministic. The issues analyzed in our paper are unrelated to such problems, and instead concern fundamental properties of IB that arise when input->output mapping is deterministic, and which occur even when the output takes a finite set of values, and all MI terms are bounded, as in classification. We have added some text to the Introduction to highlight this difference.

---

### Public Comment · (anonymous) · 2018-11-08
**On the use of the IB and associated problems**

The paper make it sound like there is something wrong with the IB in Deep Learning. But when the IB variable is 'T = w', the weights rather than the features, these problems are not there. This is quite clear in the work of Achile and Soatto (disappointing not to see that work discussed here).

---

> ### Public Comment · (anonymous) · 2018-11-08
> **reference?**
>
> ?

---

> > ### Public Comment · (anonymous) · 2018-11-24
> > **reference**
> >
> > https://arxiv.org/abs/1706.01350
> >
> > That reference was present in the version of this paper that is on ArXiv, but was removed in this submission, strangely.

---

> ### Author Response · Authors · 2018-11-14
> **Our analysis applies more generally**
>
> Thank you for your comment. Our analysis is not exclusive to deep learning but actually applies to any use of IB where the output variable is finite-valued and a deterministic function of the input, regardless of the choice of the bottleneck variable. Unfortunately, for space reasons, we are unable to discuss all the work related to IB and machine learning.

---

### Public Comment · (anonymous) · 2018-11-24
**"caveats" well known but not relevant to any real application of the IB. Missing reference to relevant literature**

For Deep Learning, it is well understood that the IB framework relies on the stochasticity of the weights during training. Tishby discussed the concept at length in his papers when talking about diffusion process of the weights and the noise in SGD. A lot of other works that address related issues, which makes this paper moot for DL, is ignored by the authors despite being 2 pages longer than the limit: Work using PAC-Bayes (https://arxiv.org/abs/1703.11008), Bayesian optimization (https://arxiv.org/abs/1506.02557, https://openreview.net/pdf?id=BJij4yg0Z), Information Theory (https://arxiv.org/abs/1706.01350), Optimization (https://arxiv.org/pdf/1803.06959.pdf), relies on information or noise in the weights to avoid the issue and still obtain results on the corresponding representations as proved by https://arxiv.org/abs/1706.01350.

In the deterministic case, T should contain no more than log(n_classes) bit for a naive application of the IBL to even make sense. This is clearly not the case on any modern deep network, where the information in each layer would be substantially larger. Even in the non-deterministic case it does not make sense if H(T) > log(size of dataset) (the representation T could be an hash of the input image), which is often the case in any problem of practical relevance. Therefore, the pathologies or “caveats” described by the authors do not show up in any realistic problems in DL; to see them, one has to build artificial examples, like those in this paper, that have no bearings in reality.

Among the other references the author miss, in the discussion of "research [that] investigates neural network training algorithms that optimize the IB Lagrangian’’: before Kolchinsky et al., 2017 and Alemi et al., 2016 there was Achille and Soatto, https://arxiv.org/pdf/1611.01353 which does the same; also relevant in that vein https://openreview.net/pdf?id=Sy2fzU9gl, which successfully applies that theory in non trivial deep learning scenarios.

---

> ### Author Response · Authors · 2018-11-26
> **Response**
>
> Thanks for your comment. The caveats discussed in our paper concern theoretical properties of IB-optimal variables and the IB Lagrangian, and hold whenever the output is a deterministic function of the input. In a machine learning context, they apply independently of the training algorithm and how the weights are produced, and in particular even when the weights are stochastic, as in SGD (since IB optimal representations do not depend on the training algorithm). Moreover, our results are also applicable to the use of IB outside of the realm of machine learning, where the concepts of "training algorithm" and "weights" may not apply.
>
> We disagree that the caveats we discuss are well known, since they have been reported in no prior literature, including the articles mentioned by the commenter. We also note that the caveats are derived analytically, and then  demonstrated on the example of MNIST, which though simple is not artificial.
>
> We appreciate the pointer to Achille and Soatto’s "Information dropout", which also proposes an optimizable approximation to the IB Lagrangian, and which we now cite. We agree that recent work on the connection between VAEs and IB, such as the mentioned https://openreview.net/pdf?id=Sy2fzU9gl, is very interesting, and some of our results may have implications there. However, the interpretation of autoencoding from the point of view of IB is quite different from that of supervised learning, and for space reasons we concern ourselves only with the latter.

---

### Meta-Review · Area_Chair1 · 2018-12-13
**Important cautions about the information bottleneck in typical learning settings**

**Confidence:** 4
**Recommendation:** Accept (Poster)

**Metareview:**

This paper considers the information bottleneck Lagrangian as a tool for studying deep networks in the common case of supervised learning (predicting label Y from features X) with a deterministic model, and identifies a number of troublesome issues. (1) The information bottleneck curve cannot be recovered by optimizing the Lagrangian for different values of β because in the deterministic case, the IB curve is piecewise linear, not strictly concave. (2) Uninteresting representations can lie on the IB curve, so information bottleneck optimality does not imply that a representation is useful. (3) In a multilayer model with a low probability of error, the only tradeoff that successive layers can make between compression and prediction is that deeper layers may compress more. Experiments on MNIST illustrate these issues, and supplementary material shows that these issues also apply to the deterministic information bottleneck and to stochastic models that are nearly deterministic. There was a substantial degree of disagreement between the reviewers of this paper. One reviewer (R3) suggested that all the conclusions of the paper are the consequence of P(X,Y) being degenerate. The authors responded to this criticism in their response and revision quite effectively, in the opinion of the AC. Because R3 failed to participate in the discussion, this review has been discounted in the final decision. The other two reviewers were considerably more positive about the paper, with one (R1) having basically no criticisms and the other (R2) expression some doubts about the novelty of the observations being made in the paper and their importance for practical machine learning scenarios.  Following the revision and discussion, R2 expressed general satisfaction with the paper, so the AC is recommending acceptance. The AC thinks that the final paper would be clearer if the authors were to carefully distinguish between ground-truth labels used in training and the labels estimated by the model for a given input.  At the moment, the symbol Y appears to be overloaded, standing for both.  Perhaps the authors should place a hat over Y when it is standing for estimated labels?